

# A low-cost mobile multidisciplinary measurement platform for monitoring geophysical parameters

Olivier F.C. den Ouden[1,2], Jelle D. Assink[1], Cornelis D. Oudshoorn[3] Dominique Filippi[4] and Läslo G. Evers[1,2]
[1]R&D Department of Seismology and Acoustics, Royal Netherlands Meteorological Institute, De Bilt, The Netherlands
[2]Dept. of Geoscience and Engineering, Delft University of Technology, Delft, The Netherlands
[3]R&D Department of Observations and Data Technology, Royal Netherlands Meteorological Institute, De Bilt, The Netherlands
[4]Sextant Technology Inc., Marton, New-Zealand
**Abstract**
Geophysical studies and real-time monitoring of natural hazards, such as volcanic eruptions or severe
weather events, benefit from the joint analysis of multiple geophysical parameters. However, typical
geophysical measurement platforms still provide logging solutions for a single parameter, due to different
community standards and the higher cost rate per added sensor.
In this work, the 'infrasound-logger' is presented, which has been designed as a low-cost mobile mul-
tidisciplinary measurement platform for geophysical monitoring. The platform monitors in particular
infrasound, but concurrently measures barometric pressure, accelerations, wind flow and uses the Global
Positioning System (GPS) for positioning of the platform. Due to its digital design, the sensor platform
can readily be integrated with existing geophysical data infrastructures and be embedded in the analysis of
geophysical data. The small dimensions and lower cost price per unit allow for unconventional experimen-
tal designs, for example high density spatial sampling or deployment on moving measurement platforms.
Moreover, such deployments can complement existing high-fidelity geophysical sensor networks. The
platform is designed using digital Micro-electromechanical Systems (MEMS) sensors that are embedded
on a Printed Circuit Board (PCB). The MEMS sensors on the PCB are: a GPS, a three-component
accelerometer, a barometric pressure sensor, an anemometer and a differential pressure sensor. A pro-
grammable microcontroller unit controls the sampling frequency of the sensors, and the data storage. A
waterproof casing is used to protect the mobile platform against the weather. The casing is created with
a stereolithography (SLA) Formlabs 3D printer, using durable resin.
Thanks to a low power consumption (9 Wh over 25 days), the system can be powered by a battery or
solar panel. Besides the description of the platform design, we discuss the calibration and performance
of the individual sensors.



# 1 Introduction

Real-time monitoring of natural hazards, such as volcanic eruptions or severe weather events benefit from the joint analysis of multiple geophysical parameters. However, geophysical measurement platforms are typically designed for the measurement of a single parameter, due to different community standards and the higher cost rate per added sensor. The quality and robustness of geophysical measuring equipment generally scales with price, due to higher material costs and research and development (R&D) expenses of the manufacturer. In addition, the deployment of such equipment comes with complex deployment and calibration procedures, and requires the presence of a robust power and data infrastructure.

Geophysical institutes often place multiple sensor platforms co-located. Meteorological institutes, for example, measure various meteorological parameters for comparison, which improves the weather observations, as well as weather forecast models. The Comprehensive Nuclear-Test-Ban Treaty Organization (CTBTO) performs various geophysical measurements at its measurement sites where possible. The International Monitoring System (IMS), which is in place for the verification of the CTBT, performs continuous seismic, hydroacoustic, infrasonic and radionuclide measurements [Marty, 2019]. In addition, the IMS infrasound arrays and radionuclide facilities host auxiliary meteorological equipment, as this data facilitates the review of the primary IMS data streams. Besides its use for verifying the CTBT, it has also been shown that a multi-instrumental observation observational network such as the IMS can provide useful information on the vertical dynamic structure of the middle and upper atmosphere, in particular when paired with complementary upper atmospheric remote sensing techniques such as lidar [Blanc et al., 2018]. Other studies that involve the analysis of multiple geophysical parameters include seismo-acoustic analyses of explosions ([Assink et al., 2018] [Averbuch et al., 2020]), earthquakes ([Shani-Kadmiel et al., 2018]), and volcanoes ([Green et al., 2012]).

National Weather Services, such as the Royal Netherlands Meteorological Institute (KNMI), have expressed an interest in measuring weather on a local scale to inform citizens and warn in case of extreme weather. In addition, such measurements allow for higher-resolution measurements of sub-grid scale atmospheric dynamics, which will contribute the improvement of short-term and now-casting weather forecasts. Therefore it became part of a low-cost citizen weather station program, to increase the spatial resolution of conventional



numerical weather prediction models. In the Netherlands, over 300 of those weather stations are contribut-
ing to a global citizen science project, Weather Observations Website (WOW)[Garcia-Marti et al., 2019]
[Cornes et al., 2020]. Nonetheless, due to the required infrastructure of the equipment, many platforms are
spatially static. Having a low-cost multidisciplinary mobile sensor platform allows for high-resolution spatial
sampling and complement existing high-fidelity geophysical sensor networks [Poler et al., 2020].
Various disciplines are applying new sensor technology to obtain higher spatial and temporal resolution
[D'Alessandro et al., 2014] for geophysical hazard monitoring. Micro-electromechanical systems (MEMS)
are small single-chip sensors that combine electrical and mechanical components and have a low energy con-
sumption. The seismic community has created low-cost reliable MEMS accelerometers [Homeijer et al., 2011]
[Milligan et al., 2011] [Zou et al., 2014] to detect strong accelerations that exceed values due to Earth's grav-
ity field [Speller and Yu, 2004] [Laine and Mougenot, 2007] [Homeijer et al., 2014]. Moreover, the infrasound
[Marcillo et al., 2012] [Anderson et al., 2018], as well as the meteorological community are integrating MEMS
sensors into the existing sensor network [Huang et al., 2003] [Fang et al., 2010] [Ma et al., 2011].
In this work, the 'infrasound-logger' is presented, which has been designed as a low-cost mobile multidis-
ciplinary measurement platform for geophysical monitoring, in particular, infrasound. The platform uses
various digital MEMS sensors, which are embedded on a Printed Circuit Board (PCB). A programmable
microcontroller unit, as well embedded on the PCB, controls the sampling frequency of the sensors and
establishes the energy supply for the sensors as well as the data-communication and storage. A waterproof
casing protect the mobile platform against weather. The casing is created with a stereo-lithography (SLA)
Formlabs 3D printer, using durable resin. Because of it's low power consumption, the system can be powered
by a battery or solar panel.
The ability to detect infrasonic signals of interest depends on the strength of the signal relative to the noise
levels at the receiver side, the signal to noise ratio (SNR). The signal strength depends on the transmission
loss that a signal experiences propagating from source to receiver. Infrasound measurements benefits from
insights in the atmospheric noise levels (e.g., wind conditions), the meteorological conditions (e.g., baro-
metric pressure, temperature, and humidity), as well as the movement and positioning of the sensors (e.g.,
accelerations) [Evers, 2008].



While there are clear benefits associated with a MEMS-based mobile platform (e.g., cheap and rapid de-
ployments to (temporarily) increase coverage), MEMS sensors are known to be less accurate than con-
ventional high-fidelity equipment. Especially digital MEMS sensors, which have a build-in Analog-Digital-
Converter (ADC), are known for their high self-noise level. Nonetheless, they could be used near a geo-
physical sources which generate high SNR signals. Several geophysical measurements [Marcillo et al., 2012]
[Grangeon and Lesage, 2019] [Laine and Mougenot, 2007] [D'Alessandro et al., 2014] show the benefit of MEMS
sensors, and how they complement the existing sensor network.
In this paper, the design and calibration of the 'infrasound-logger' is discussed. Due to its digital design, the
platform can readily be integrated in existing geophysical sensor infrastructures. The remainder of this article
is organized as follows. Section 2 introduces the mobile platform, its design and features. Section 3 describes
the various sensor that are embedded on the platform as well as the relative calibrations with high-fidelity
reference equipment. Firstly, a novel miniature digital infrasound sensor is introduced and its theoretical
response is derived. Secondly, the barometric MEMS sensor is discussed. A novel wind sensor which relies on
thermo-resistive elements is discussed next, followed by a discussion of the on-board MEMS accelerometer.
In Section 4 the overall performance and design of the platform are discussed and summarized, from which
the conclusions are drawn.

## 99    2    Mobile platform design

### 100    2.1    Circuit design

The mobile platform contains a PCB, which been created to embed the MEMS sensors and to facilitate
the electrical circuits. The PCB carries a Digital Low Voltage Range (DLVR) differential pressure sensor,
an anemometer, as well as an accelerometer and barometeric pressure sensor, in addition to a GPS for
location and timing purposes (Figure 1-a). The sensors are controlled by a MSP430 microcontroller, which is
integrated on the PCB, and are powered by a 1800 mAh lithium battery. Protecting the PCB is done with a
weather- and waterproof casing, which has been designed (Figure 1-b) with the dimensions 110mm x 38mm
x 15mm.



The communication between microcontroller and MEMS sensor on the PCB is either be done by Inter-
Integrated Circuit (I2C) or Serial Peripheral Interface (SPI). Both communication methods are bus protocols
and allow for serial data transfer. However, SPI handles full-duplex communication, simultaneous communi-
cation between microcontroller and MEMS sensor, while I2C is half-duplex. Therefore, I2C has the option of
clock stretching, the communication is stopped whenever the MEMS sensor is not able to send data. Besides,
I2C has build-in features to verify the data communication (e.g., start/stop bite, acknowledgement of data).
Although the I2C protocol is favourable, it requires more power. Furthermore, the microcontroller handles
the data storage. The PCB includes a 64 mb flash memory, which is used to store the data. The raw output
of the digital MEMS sensors are stored as bits.

## 2.2 Casing design for pressure measurements

The mobile sensor platform is designed to measure atmospheric parameters. Hence, a waterproof casing has
been created, by a Formlabs SLA 3D printer [Formlabs, 2020], to protect the PCB. Because of the use of
a Durable Resin, the casing is waterproof and air-tight. At the bottom of the casing, a dome structure is
integrated (Figure 1-c), which acts as an inlet to both the absolute and differential pressure sensors. Note
that the dome is not connected to the inside of the casing. The inlets of both sensors and a capillary are
integrated in the dome designs, and sealed with silicon glue, avoiding water and air leakage. Moreover, a
Gore-TEX air-vent sticker [Gore-Tex, 2020] is used to cover the dome, which allows airflow but restrains
water and salt in case of measurement near or above the ocean.
By this design, the volume of the casing acts as a backing volume. One inlet of the differential pressure sensor
is attached to the outside (via the dome) while the casing encloses the other inlet. A PEEKsil$^{TM}$ Red series
capillary is attached to the outside of the casing, ensuring pressure leakage between the backing volume and
the atmosphere.

## 2.3 GPS

For measuring geophysical parameters on a high-resolution temporal scale, it is crucial to know the position
and time of the measurement at high precision. To maintain knowledge regarding the position, a GNS2301



<sub>133</sub> GPS is mounted on the PCB [Texim Europe, 2013]. The GPS has a spatial accuracy of ± 2.5 m.

<sub>134</sub> Besides providing an accurate position, the GPS also prevents drifting of the internal clock of the microcon-

<sub>135</sub> troller under the influence of, for example, weather. The time root mean square jitter, the deviation between

<sub>136</sub> GPS and true time, is ± 30 nanoseconds.

## 3   Sensor descriptions

### 3.1   Infrasound sensor

<sub>139</sub> The human audible sound spectrum is approximately between 20 to 20,000 Hz. Frequencies below 20 Hz

<sub>140</sub> or above 20 kHz are referred to as infrasound and ultrasound, respectively. The movement of large air

<sub>141</sub> volumes generates infrasound signals with amplitudes in the range of millipascals to tens of pascals. Ex-

<sub>142</sub> amples of infrasound sources include earthquakes, lightning, meteors, nuclear explosions, interfering oceanic

<sub>143</sub> waves and surf [Campus and Christie, 2010]. Detection of infrasound depends on the strength of the signal

<sub>144</sub> relative to the noise levels at a remote sensor (array), i.e., the signal-to-noise ratio. The signal strength

<sub>145</sub> depends, in turn, on the transmission loss that a signal experiences, while propagating from source to re-

<sub>146</sub> ceiver [Waxler and Assink, 2019]. The noise are predominantly determined by local wind noise conditions

<sub>147</sub> [Raspet et al., 2019], in addition to the sensor self-noise. Due to the presence of atmospheric waveguides

<sub>148</sub> and low absorption at infrasonic frequency [Sutherland and Bass, 2004], infrasonic signals can be detected

<sub>149</sub> at long distances from an infrasonic source. Assumed that the source levels are sufficiently high so that the

<sub>150</sub> long-range signal is above the ambient noise conditions on the receiver side, and the sensor is sensitive enough

<sub>151</sub> to detect the signal.

<sub>152</sub> The infrasonic wavefield is conventionally measured with pressure transducers since such scalar measurements

<sub>153</sub> are relatively easy to perform. Those measurements can either be performed by absolute or differential

<sub>154</sub> pressure sensors. An absolute pressure sensor consists of a sealed aneroid and a measuring cavity, which

<sub>155</sub> is connected to the atmosphere. A pressure difference within the measuring cavity will deflect the aneroid

<sub>156</sub> capsule. The mechanical deflection is converted to a voltage [Haak and De Wilde, 1996]. The measurement

<sub>157</sub> principle of a differential infrasound sensors relies on the deflection of a compliant diaphragm, which is





---

mounted on a cavity inside the sensor. The membrane deflects due to a pressure difference inside and
outside the microphone, which occurs when a sound wave passes. A pressure equalization vent is part of the
design to make the microphone insensitive to slowly varying pressure differences originating from long period
changes in weather conditions [Ponceau and Bosca, 2010]. Acoustic particle velocity sensors constitute a
fundamentally different class of sensors, that measure the airflow over sets of heated wires. This information
quantifies the 3-D particle velocity at one location, since the measurement is carried out in three directions
[De Bree et al., 2003] [Evers and Haak, 2000]. Although the design of such sensors is more involved and the
sensors are far more costly, these sensors do allow for the measurement of sound directivity at one position,
besides just the loudness.
Various studies show that the sensitivity curves of infrasound sensors [Ponceau and Bosca, 2010] [Merchant, 2015]
[Slad and Merchant, 2016] [Marty, 2019] [Nief et al., 2019] lie below the global low ambient noise curves
[Brown et al., 2014], which have been generated from global infrasound measurements using the IMS. Typical
infrasound sensor networks, such as the IMS, make use of analog sensors that are connected to a separate
data logger to convert the measured voltage differences to a digital signal. The characteristic sensitivity of the
sensor determines the sensor resolution, i.e., the smallest difference that can be detected by the sensor. The
resolution of the built-in analog-to-digital converters (ADC) and the digitizing voltage range determine the
resolution of the datalogger. Current state-of-the-art dataloggers have a 24-bit resolution. New infrasound
sensor techniques involve digital outputs, since the ADC conversion is realized inside the sensor.

### 3.1.1 Sensor design

In this section, the design of the mobile digital infrasound sensor is discussed, the KNMI mini-microbarometer
(mini-MB). The design of this instrument is based on the following requirements. The sensor should have a
flat, linear, response over a wide infrasonic frequency band, e.g., 0.05 - 10 Hz. The sensor should be sensitive
to the range of pressure perturbations that occur in this frequency band, which are in the range of millipascals
to tens of pascals. Moreover, the self-noise of both the sensor and logging components should be below the
ambient noise levels of the IMS [Brown et al., 2014]. Taking this into account, the sensor requires as well to
be low-cost (i.e., tens of dollars), small in dimensions (i.e., millimeter), and have a low energy consumption





(i.e., milliampere) .
In this study, infrasound is measured with a differential pressure sensor. The measurement principle relies
on the deflection of a diaphragm, which is mounted between two inlets. One inlet is connected to the
atmosphere while the other is connected to a cavity (Figure 2). The digital MEMS DLVR-F50D differential
pressure sensor from All Sensors Inc [DLV, 2019] is used as sensing element within the mini-MB. This sensor
has a dimension of 16.5mm x 13.0mm x 7.3mm and has a linear response between $\pm$ 125 Pa with a maximum
error band of $\pm$0.7 Pa. A Wheatstone bridge senses the deflection of the diaphragm by measuring the changes
in the piezo-resistive elements attached to the diaphragm. The output of the sensor is an analog voltage,
that is subsequently digitized by the built-in 14-bit ADC, offering a maximum resolution of 0.02 Pa/count.

### 193 3.1.2 Theoretical response

To measure differential pressure, the atmosphere is sampled through inlet A, which has a low resistance
$(R_1)$, and is connected to a small fore-volume $(V_1)$. Inlet B is connected to a backing volume $(V_2)$, which is
connected to the atmosphere by capillary that acts as a high acoustic resistance $(R_2)$, which determines the
low-frequency cut off. Due to an external pressure wave, an observed pressure difference between the two
inlets occurs and causes a deflection of the membrane $(C_d)$ (Figure 2-a).
A theoretical response, $D(i\omega)$ for a differential pressure sensor, as function of the angular frequency $\omega(= 2\pi f)$,
has been derived by [Mentink and Evers, 2011] following [Burridge, 1971]:

$$D(i\omega) = \frac{i\omega\tau_2}{1 + i\omega\tau_2 A + (i\omega)^2 \tau_1 \tau_2 B} \tag{1}$$

where,

$$A = 1 + \frac{\tau_1}{\tau_2} + \frac{R_1}{R_2} + \frac{C_d}{C_2}, \quad B = 1 + C_d(\frac{1}{C_1} + \frac{1}{C_2}) \tag{2}$$

$$\tau_j = R_j C_j, \quad C_j = \frac{V_j}{P_{\text{atm}}\gamma} \tag{3}$$

and $P_{\text{atm}}$ indicates the ambient barometric pressure, and $\gamma$ is the thermal conduction of air. $\tau_j$ represent
the time constants, and depend on $R_1$, and $R_2$, which are the resistances of the inlet and capillary, and





$C_1$, and $C_2$, the capacities of the fore and backing volume.

| KNMI mini-MB sensor specifications | | | |
|---|---|---|---|
| **Components** | | **Conditions** | |
| Inlet length | $l_1 = 3\mathrm{x}10^{-2}\mathrm{m}$ | Ambient pressure | $P_\mathrm{atm} = 101\mathrm{x}10^3\mathrm{Pa}$ |
| Inlet diameter | $a_1 = 2\mathrm{x}10^{-2}\mathrm{m}$ | Isothermal gas constant | $\gamma_{iso} = 1$ |
| Capillary length | $l_2 = 5\mathrm{x}10^{-2}\mathrm{m}$ | Adiabatic gas constant | $\gamma_{adi} = 1.403$ |
| Capillary diameter | $a_2 = 1\mathrm{x}10^{-4}\mathrm{m}$ | Thermal conductivity | $\kappa = 2.5\mathrm{x}10^{-2}\ \mathrm{W\ m^{-1}\ K^{-1}}$ |
| Diaphragm sensitivity | $C_d = 7.5\mathrm{x}10^{-11}\mathrm{m^4 s^2 kg^{-1}}$ | Heat capacity | $\rho\,\mathrm{c}_p = 1.1\mathrm{x}10^3\ \mathrm{J\ m^{-3}\ K^{-1}}$ |
| **Parameters** | | | |
| Inlet resistance | $R_1 = 8.7\mathrm{x}10^3\ \mathrm{kg\ m^{-4}\ s^{-1}}$ | Fore volume | $V_1 = 4.5\mathrm{x}10^{-7}\ \mathrm{m}^3$ |
| Capillary resistance | $R_2 = 2.3\mathrm{x}10^{10}\ \mathrm{kg\ m^{-4}\ s^{-1}}$ | Backing volume | $V_2 = 16.5\mathrm{x}10^{-6}\ \mathrm{m}^3$ |
| Size fore volume | $L_1 = 2\mathrm{x}10^{-4}\mathrm{m}$ | Size backing volume | $L_2 = 4\mathrm{x}10^{-4}\mathrm{m}$ |

Table 1: KNMI mini-MB components, parameter values and standard conditions used in the computations.

Figure 2-a represents the sensor setup from an acoustical perspective, where Figure 2-b represents the elec-
trical analogs of the sensor. The acoustical pressure difference $(p' = p'_1 - p'_2)$ and volume flux $(f')$ are
interpreted as an electrical voltage $(U = U_1 - U_2)$ and current $(I)$. The equivalent of the electrical resistance
$(R)$ corresponds to the ratio between acoustical pressure and the volume flux, whereas the capacitance $(C)$
relates to the ratio of volume and ambient barometric pressure. The mechanical sensitivity of the diaphragm
$(C_d)$ is the ratio of volume change and pressure change [Zirpel et al., 1978].
From an analysis of Eq. 1, it follows that inlet A dominates in the high-frequency limit. Hence, $1/2\pi\tau_1$
indicates the high-frequency cut-off of the sensor:

$$\lim_{\omega \to +\infty} D(i\omega) \sim \frac{1}{i\omega\tau_1 B} = \frac{1}{\frac{i\omega R_1 V_1}{P_\mathrm{atm}}\left(1 + C_d\left(\frac{P_\mathrm{atm}}{V_1} + \frac{P_\mathrm{atm}}{V_2}\right)\right)} \tag{4}$$

While at low frequencies it is obtained that frequencies much smaller than $1/\tau_2$ are averaged out. Therefore
the low-frequency limit can be determined as:



$$\lim_{\omega \to 0} D(i\omega) \sim i\omega = \frac{i\omega R_2 V_2}{P_{\text{atm}}} \tag{5}$$

which is controlled by the characteristics of the capillary, $R_2$, and the size of the backing volume, $V_2$. The
acoustical resistance of the inlet $R_1$ and the capillary $R_2$ is described by using Poiseuille's law [Washburn, 1921],
which couples the resistance of airflow through a pipe (i.e., an inlet or capillary) to its length $l_j$ and diameter
$a_j$, by:

$$R_j = \frac{8 l_j \eta}{\pi a_j^4} \tag{6}$$

Where $\eta$ stands for the viscosity of air, which equals 18.27 $\mu$Pa·s at 18°C. Combining Equations 5 and 6
results in the theoretical low-frequency cut-off:

$$f_l \sim \frac{P_{\text{atm}}}{2\pi R_2 V_2} \tag{7}$$

Besides the high and low ends of the response, it is of interest to determine the sensor response behavior
within the passband $((\tau_2^{-1} < \omega < \tau_1^{-1}))$.

$$D(i\omega) \sim (\tau_2^{-1} < \omega < \tau_1^{-1}) = \frac{1}{1 + \underbrace{\tau_1/\tau_2}_{1} + \underbrace{R_1/R_2}_{2} + \underbrace{C_d/C_2}_{3}} \tag{8}$$

The three contributions in the denominator influence the passband behavior of the sensor:
1. A broadband frequency response depends on a constant pressure within the reference volume over the

225       frequencies of interest (i.e., $\tau_1 \ll \tau_2$)

2. The pressure difference at the diaphragm is determined by the relative acoustical resistances that are

227       connected to the sensor. The stability of the sensor response is assured by the large resistance of the

228       capillary, because of which $R_1 \ll R_2$.

3. The sensor response depends on the ratio between the volumetric displacement of the diaphragm ($C_d$)

230       versus the reference volume ($C_2$). For the mini-MB, this term can be neglected.





Figure 3 shows the theoretical sensor frequency response for amplitude (Fig. 3-a) and phase (Fig. 3-b) for
isothermal (red) and adiabatic (blue) behavior. The transitional behavior of the sensor response between
isothermal and adiabatic behavior will be discussed in the next section.

### 3.1.3  Adiabatic-Isothermal transition

Due to the presence of heat conduction within the sensor, the compressive behavior of air is neither isothermal
nor adiabatic. Instead, a transition from isothermal to adiabatic behavior is expected in the infrasonic
frequency band [Richiardone, 1993] [Mentink and Evers, 2011]. In the transition zone, the heat capacity
ratio can be effectively described by:

$$\overline{\gamma} = \Lambda\gamma \tag{9}$$

where $\Lambda$ indicates the correction factor, to heat capacity ratio $\gamma$. A difference in $\Lambda$ will influence the capaci-
tance values of the fore and backing volumes (Eq. 3).
Whether a sound wave in an enclusure behaves isothermally or adiabatically depends on the size of the
thermal penetration depth $\delta_t$ relative to characteristic length $L$ of the enclosure. $L$ is defined as the ratio
between the volume and surface of the enclosure, i.e. $L = \frac{V}{S}$. The thermal penetration depth is specified as
the gas layer thickness in which heat can diffuse through, during the time of one wave period and is derived
as $\delta_t = \sqrt{\frac{2\alpha}{\omega}}$. Where $\alpha = \frac{\kappa}{\rho c_p}$ indicates the thermal diffusivity, defined as ratio of thermal conductivity ($\kappa$)
and heat capacity per unit volume ($\rho c_p$). Adiabatic gas behaviour is obtained when $\frac{\delta_t}{L} \ll 1$, isothermal gas
behaviour when $\frac{\delta_t}{L} \gg 1$. The correction factor $\Lambda$ is a function of $\delta_t/L$, and is thus frequency dependent,
which can be derived as:

$$|\Lambda| = \sqrt{X^2 + Y^2}, \quad \arg(\Lambda) = \frac{\pi}{2} + \arctan(\frac{X}{Y}) \tag{10}$$

where

$$X = x(\gamma_{adi} - 1) - \gamma_{adi}, \quad Y = y(\gamma_{adi} - 1) \tag{11}$$





$x$ and $y$ represent the real and imaginary components of a complex-valued function $Z(\frac{\delta_t}{L})$, which is dependent
on the geometrical shape of the enclosure and the thermal pentration depth. In between the adiabatic and
isothermal limits, the correction factor $\Lambda$ describes the transition from an adiabatic heat ratio (i.e., $\gamma = 1.4$)
to an isothermal heat ratio, i.e. $\gamma = 1$. The transition frequency $\bar{f}$ defines the point where the maximum
correction of $\Lambda$ occurs, i.e., for which $L\delta_t \approx 1$, from which follows that $\bar{f} = \frac{\alpha}{\pi L^2}$.
In case of the mini-MB the fore and backing volume have different shapes and sizes. The backing volume can
be described as a long cylinder, $L_2$, whereas the fore volume has the shape of a rectangular, $L_1$. According to
those geometries, the transition frequency $\overline{f}$ of the fore and backing volume are 0.5 and 2.2 Hz, respectively.
Since $\overline{f}_1 \cdot \tau_1 \ll 1$ and $\overline{f}_2 \cdot \tau_2 \gg 1$ the sensor response above $\tau_1^{-1}$ is adiabatic, while the response below $\tau_2^{-1}$
is isothermal. Therefore the main effect of the thermal conduction correction is found to be in the passband
region (Eq. 8).
The mini-MB has been designed to have a broadband response, therefore only the third term of the dominator
is influenced by the correction factor. The effect of thermal conduction to the response is due to ratio $\frac{C_d}{C_2}$,
which means that the correction factor is characterized by the geometric component of the backing volume.

$$Z(\frac{\delta_t}{L}) = 1 - \frac{2J_1(\zeta)}{\zeta J_0(\zeta)} \qquad (12)$$

here Z indicates the characteristic correction assuming a long cylinder [Mentink and Evers, 2011]. $\zeta =$
$\sqrt{-2i}\frac{L}{\delta_t}$ indicates the ratio of $L$ to $\delta_t$, while $J_0$ and $J_1$ are zeroth and first order Bessel functions of the first
kind.
The corrected theoretical sensor response is obtained by substituting $\overline{C_j} = \frac{C_j}{\Lambda}$. Figure 3-c shows the value of
$\overline{\gamma}$ in the transaction zone between isothermal and adiabatic gas behaviour. The black line in Figure 3-a and
b indicates the corrected theoretical sensor response.
In the case of the mini-MB the isothermal-to-adiabatic transition results in an effect on the amplitude of
$\Delta|D| = (\gamma - 1)\frac{C_d}{C_2} = 2.8\%$ and on the phase of less than a degree. Note that $\frac{C_d}{C_2} \ll 1$ implies that the
backing volume is relatively large such that the change in gas behavior does not influence the sensitivity of
the diaphragm.





### 3.1.4 Gore-Tex air-vent

As discussed in Section 3.1.2., the high and low-frequency cut-off are controlled by the resistivity of the inlet and backing volume, respectively. A Gore-Tex V9 sticker is added to the opening of the pressure dome of the casing, which changes the resistivity of the inlets. The Gore-Tex V9 vent allows an airflow of $2\text{x}10^{-8}\text{m}^3\text{s}^{-1}\text{m}^{-2}$. Poiseuille's second law, Equation 6, shows the airflow resistivity caused by an open pipe, and can be re-written as;

$$R_j = \frac{\Delta p}{q_v} \qquad (13)$$

where $\Delta p$ indicates the pressure difference between both sides of the pipe, and $q_v$ the volumetric airflow.

For the differential pressures that the mini-MB sensor is able to sense, ranging from 0.02 to 125 Pa, with a Gore-Tex air-vent area of $5\text{x}10^{-2}$ m$^2$, the equivalent resistivity $R_{\text{gore}}$ is ranging from $5\text{x}10^5$ to $3.125\text{x}10^8\text{kgm}^{-4}\text{s}^{-1}$. Comparing the resistivity of the air-vent with the resistivity values of the capillary and the sensors inlet, Table 1, it follows that only the resistivity of the inlet will be influenced by the air-vent. Assuming the vent behaves linear, the high frequency cut-off of the sensor decreases to a value of around 15 Hz. Figure 3 shows the theoretical transfer function for the mini-MB with a Gore-Tex air-vent attached to the inlet. The high frequency cut-off is shifting between the dotted line and the dashed line, due to varying values of $R_{\text{gore}}$.

### 3.1.5 Experimental response

The theoretical sensor response describes the high and low-frequency cut-off. From Eq. 7 and the parameters listed in Table 1, it follows that the mini-MB has a theoretical low-frequency cut-off of 0.042Hz. A sudden over or under pressure (i.e., impulse response) is applied to the sensor to determine the low-frequency cut-off experimentally[Evers and Haak, 2000]. The impulse forces the diaphragm out of equilibrium. The capillary and the size of the backing volume control the time to return into equilibrium again. The time it takes for the diaphragm to reach equilibrium again corresponds to a characteristic relaxation time that is proportional to the low-frequency cut-off.





The outcome of the experimental low-frequency cut-off was determined to be 0.044±0.0025Hz. The theoretical
low frequency cut-off falls within the error margins of the experimental cut-off frequency. Small difference
between both are assumed to be due to experimental errors in timing the relaxation time as well as small
imperfections in the used capillary [Evers, 2008]. It follows from Eq. 6 that the low-frequency cut-off is
inversely proportional to the radius to the fourth power. Hence, a one percent deviation in the capillary
radius will lead to a four percent deviation in low-frequency cut-off.
**3.1.6 Sensor self-noise**
The resolution, the smallest change detectable by a sensor, depends on the sensor measurement range and
the number of ADC bits. Having a linear response over a pressure range of ± 125 Pa and a 14-bit build-in
ADC results in a resolution of 0.02 Pa/count. The accuracy of the measurement depends, besides the ADC
resolution, on the internal error of the sensor, the self-noise. The self-noise corresponds to the deformation of
the diaphragm caused by the mass of the diaphragm plus the electrical noise from the digitiser. As it is a dig-
ital sensor, it is not possible to follow the conventional methods to determine self-noise [Sleeman et al., 2006].
Therefore the self-noise is determined by opening both inlets to a closed pressure chamber, ensuring no pres-
sure difference between both inlets. The outcome stated that the self-noise falls within the maximum error
band of the sensor, ±0.7 Pa [DLV, 2019]. Since no backing volume is used, and the cavities at both sides of
the diaphragm are small, the relation $\frac{C_d}{C_2}$ changes (Eq. 8). Due to this, it is necessary to correct the sensor
response for the adiabatic to isothermal transition. (Section 3.1.3).
The consistency of the self-noise is determined by calculating the Power Spectral Density (PSD) curves for
each hour over a test period of 24 hours [Merchant and Hart, 2011]. Figure 4-a shows in black the average
90 percentile confidence interval of the self-noise. Note that the instrumental self-noise exceeds the global
low noise model [Brown et al., 2014] at frequencies above 0.4 Hz. Compared to high-fidelity equipment
that typically fall completely below the global low noise models, such self-noise levels are relatively high,
yet comparable to levels that are attained by similar sensor designs [Marcillo et al., 2012]. Furthermore,
note that the self-noise follows the dynamic range of a 12-bit ADC, as indicated by the gray dotted line
[Sleeman et al., 2006]. The sensor has a maximum 'no missing code' of 12-bits, the effective number of bits





[DLV, 2019].

### 3.1.7  Sensor comparison

A comparison between the mini-MB and a Hyperion IFS-5111 sensor [Merchant, 2015] is made to assess the performance of the mini-MB relative to the reference Hyperion sensor. Both sensors have been placed inside a cabin next to the outside sensor test facility at the leading author's institute. There is a connection to the outside pressure field through air holes in the wall of the cabin. The Hyperion sensor has been configured with a high-frequency (HF) shroud. Figure 4-a and b show the PDF [Merchant and Hart, 2011] of the data recorded by the mini-MB and the Hyperion sensor, respectively. Both sensors resolved the characteristic microbarom peak around 0.2Hz [Christie and Campus, 2010].

A direct comparison of the pressure recordings is shown in Figures 4-c, -d, and -e. Figure 4-c shows the absolute difference in amplitude over frequency, where panel d indicates the phase difference between both sensors. Panel e shows the relative difference between the mini-MB and the Hyperion sensor. The sensors are in good agreement over the passband frequencies. A larger deviation is shown for the low end ($f < 0.07$ Hz) and high end frequencies($f > 8$ Hz). At frequencies between 0.07 and 1 Hz, the pressure values are positively biased by $5 \pm 1$ dB, which equals an error of $\pm$ 0.005 Pa (Figure 4-e). Above 1 Hz, the pressure values are biased by $10 \pm 5$ dB, which equals an error of $\pm$ 0.02 Pa.

The deviation in the low frequency spectrum is caused by the backing volume. The high frequency deviation is due to the relatively high noise level of the mini-MB. For the higher frequencies, the mini-MB PDF follows the 12-bit dynamic range. Only in case of significant events or loud ambient noise, the sensor is capable of sensing pressure perturbations in the high-frequency range. Nonetheless, over the entire frequency band the mini-MB falls within a 30 dB error range compared to the Hyperion IFS-5111 sensor.

## 3.2  Meteorological parameters

The detectability of infrasound is directly linked to wind noise conditions and the stability of the atmosphere in the surrounding of the infrasound sensor, since noise levels are increased when turbulence levels are high. Therefore, it is beneficial to have simultaneous measurements of the basic meteorological parameters, i.e.,



---

pressure, wind and temperature. The sub-sections below describe the different meteorological measurements
contained on the sensor platform.

### 3.2.1 Barometric pressure sensor

The barometric pressure is sensed by the LPS33HW sensor [STMicroelectronics, 2017], which is part of the
pressure dome. Similarly to the differential pressure sensor, piezo-resistive crystals measure the barometric
pressure.
Calibration tests are performed within a pressure chamber, in which a cycle of static pressures between 960
and 1070 hPa can be produced. Besides the MEMS sensor, the chamber is equipped with a reference sensor.
This procedure resulted in a calibration curve, which describes the pressure dependent systematic bias. After
correcting for the bias, the LPS sensor has an accuracy of ± 0.1 hPa, i.e., the LPS sensors measures values
that are within ± 0.1 hPa of the value measured by the KNMI reference sensor. Furthermore, the LPS sensor
has been field tested, along a Paroscientific Digiquartz 1015A barometer, which has an accuracy of 0.05 hPa.
From the distribution of observations, it can be estimated that the LPS sensor has a precision of ±0.1 hPa
for 93% of the time. For the remainder, the maximum deviation was ±0.15 hPa.

### 3.2.2 Wind sensor

The pressure field at infrasonic frequencies consists, in addition to coherent acoustic signals, to a large
degree of pressure perturbations due to wind and turbulence, and which is generally referred to as wind-noise
[Walker and Hedlin, 2010]. Wind noise is present over the complete infrasonic frequency range with a typical
noise amplitude level decrease with increasing frequencies, following a $f^{-5/3}$ slope [Raspet et al., 2019].
For the reduction of wind noise a Wind-Noise-Reduction-System (WNRS) can be put in place [Walker and Hedlin, 2010]
[Raspet et al., 2019]. Most WNRSs applied consist of a non-porous pipe rosette, with low impedance inlets
at the end of each pipe. All pipes are connected to four main pipes, which connect to the microbarometer.
Doing so, the atmosphere is sampled over a larger area, and thus small incoherent pressure perturbations
(e.g., wind) are filtered out.
The sensor presented in this paper is designed for mobile sampling campaigns. In such cases, the application





of similar WNRS filters cannot be attained. Not having a WNRS decreases the SNR, measuring wind with
an anemometer will give an insight into the wind conditions. Therefore, a simultaneous measurement of wind
and infrasound provides better insight into the infrasonic SNR conditions.

### Sensor design

To measure the wind conditions, a 2D omni-directional heat mass flow sensor has been designed, which is a
robust and passive anemometer (Figure 6-a). The sensor is built with a central heating element, which heats
up to approximately 80°C, and is circularly surrounded by six TDK thermistors [TDK, 2018]. Depending on
the wind direction and speed, the temperature field around the center element is modified. The wind speed
and direction can be estimated from the 2D temperature gradient, i.e., its absolute value and direction.

### Theoretical response

The six sensing elements are placed within a distance of one centimeter from the heating element, while two
thermistors and the heating element are at a spatial angle of 60°. The thermistors are used to measure the
temperature gradient that is caused by the wind flow, since the resistance is strongly sensitive to temperature.
The thermistors are made of semiconductor material and have a negative temperature coefficient. The
resistance decreases non-linearly with increasing temperature. The Steinhart-Hart equation approximately
describes the temperature $T$ as a function of resistance value $R_\Omega$ [Steinhart and Hart, 1968]:

$$\frac{1}{T} = C_{\Omega_1} + C_{\Omega_2}(\ln(R_\Omega)) + C_{\Omega_3}(\ln(R_\Omega)^3) \tag{14}$$

where $C_{\Omega_1}, C_{\Omega_2}$, and $C_{\Omega_3}$ are the thermistor constants, which can be determined by taking three calibration
measurements, for which the temperature and resistance are known, and solving the three equations simulta-
neously. Figure 6-b shows the sensitivity curve for the TDK thermistor. The thermistor has a relative value
of 1Ω at 25°$C$, and a precision of ±4%/°C, which leads to a 0.05°C error. In the next section, this error
value is placed in context by modeling the expected temperate difference under representative meteorological
conditions.

**Numerical sensor response**

The heating element needs to be able to transfer a minimum temperature difference around the sensing elements (i.e., the sensing elements error). A numerical model has been built in ANSYS to define the amount of temperature difference around the sensing elements under different meteorological circumstances. The model is a first approximation of the sensitivity and is based on homogeneous laminar airflow passing by the sensor. Turbulent flow along the anemometer causes uncertainties in wind direction and speed.

This approach follows a numerical forward modeling technique to approximate the shape of the heat probe and its intensity at a sensing element. The model was run at stable meteorological parameters (i.e., 8°C air temperature, 50% humidity, and 10 m/s wind speed). The outcome shows that under those circumstances, the sensing element experiences a temperature difference of around 4°C. Together with the outcome of the sensitivity curve of the thermistors, it is concluded that the designed sensor can resolve this airflow and is used to estimate wind speed and direction.

**Reference calibration**

Experimental calibration of the anemometer has been performed at the KNMI's calibration lab. The calibration lab features a wind tunnel, which generates a laminar airflow ranging between 0 - 20 m/s. Within the wind-tunnel, two mechanical anemometers are installed, which serve as reference sensors. The mobile platform with its MEMS anemometer is installed right below one of the reference sensors to ensure that the mobile platform does not obstruct the laminar flow in the tunnel.

The calibration procedure consists of multiple independent calibration tests that will be described next. First, the sensor is placed inside the wind tunnel while there is no airflow. This way, the relative difference between the sensing elements is determined, the so-called zero-measurement. By correcting for the relative difference, the sensor is corrected for the internal bias, which varies around $\pm$ 25 ohm. After correcting for the sensor bias, the sensor is placed at different angles with respect to the air flow. For every angle, the flow speed is varied between 0 tot 20 m/s.

The calibration shows that the measured resistance of the thermistors increases with increasing wind speeds.





---

High wind speeds increasingly cool down the thermistors, resulting in higher resistances. Figure 6-c shows
the measured resistance of the six thermistors over the wind speed. To convert resistance into wind speed, a
polynomial curve has been fitted over the average measured resistance (Fig. 6-c, black line).
The accuracy of the wind direction is determined by interpolating the measured velocities into a gradient
field over the sensor. The wind direction is obtained by calculating the mean gradient vector of this gradient
field. Three different sensor set-ups show the accuracy and precision over increasing wind speeds as a function
of directivity. The outcome of calibration set-ups 1 (270°), 2 (90°), and 3 (60°) are shown respectively in
Figure 6-d, -e, and -f. The mean direction over all wind speeds, for the three set-ups, are 93°, 265°, and 62°.
The standard deviation shows that the accuracy of the sensor is ±18°. Furthermore, it is shown that the
precision of the wind direction increases with increasing wind speeds.

## 3.3  Accelerometer

The sensing element of the infrasound sensor on this platform is a sensitive diaphragm. Strong accelerations
of the platform will cause a deflection of the diaphragm and may obscure infrasonic signal levels. In addition,
such accelerations may be interpreted erroneously as infrasound if no independent accelerometer information
is available. To be able to separate the mechanical response of the sensor from actual signals of interest, the
platform measures accelerations for which the LSM303, a 6-axis inertial measurement unit (IMU), is deployed
[STMicroelectronics, 2018]. The LSM303 consists of a 3-axis accelerometer and 3-axis magnetometer. The
measurement range of the accelerometer varies between approximately 2-16 g. The magnetometer is out of
the scope of this study, and therefore neglected for the remainder.
Accelerometers measure differential movement between the gravitational field vector and its reference frame.
In the absence of linear acceleration, the sensor measures the rotated gravitational field vector, which can be
used to calibrate the sensor. A rotational movement of the sensor will result in an acceleration. The IMU
is a digital sensor with a built-in 16-bits ADC and has a resolution of 0.06 mg when choosing the lowest
measurement range.
A comparison test has been carried out in the seismic pavilion of the author's institute. Inside this pavilion
the LSM is compared to a Streckeisen STS-2 seismometer connected to a Quanterra Q330, as a reference





sensor [KNMI, 1993]. Both sensors are installed on pillars, to ensure a good coupling between the subsurface
and the sensor. The comparison test, which is based on 24 hours of recording, shows that the accuracy of
the LSM303 3-axis accelerometer is $\pm 1.5$ mg (1.5 cm/s$^2$). Figure 7 shows the PDF's of the comparison test
for the MEMS and STS-2 sensor. While the sensors are deployed on the same seismic pillar, and are thus
subject to similar seismic noise conditions, the MEMS sensor was not able to measure ambient seismic noise
([Peterson, 1993] [McNamara and Buland, 2004]) due to its high self-noise level. The LSM accelerometer
exceeds both the U.S. Geological Survey New High Noise Model (NHNM) [Peterson, 1993] as well as the
STS-2 reference sensor by at least 35 dB.
It is therefore unlikely to use this IMU for monitoring purposes of ambient seismic noise or teleseismic events.
Previous studies drew similar conclusions concerning the performance of MEMS accelerometers. Various
calibration set-ups are considered while comparing MEMS accelerometers with conventional accelerometers
of geophones [Hons et al., 2008] [Albarbar et al., 2009] [Anthony et al., 2019], each concluding that the ac-
curacy of the MEMS is not sufficient for recording ambient seismic noise. However, strong local events or in
case of extremely noisy environments the MEMS sensor will be able to resolve those seismic signals.

## 4  Discussion and Conclusion

In this study, the constructional efforts and calibration protocols of the "infrasound-logger" are presented.
The "infrasound-logger" is a low-cost mobile multidisciplinary sensor platform for the monitoring of geophys-
ical quantities and includes sensors for the measurement of infrasound, acceleration, as well as barometric
pressure and wind.
The platform uses the newest sensor technology, i.e., digital MEMS, which have a build-in ADC. The MSP430
programable microcontroller unit controls the sampling of the ADC and the storage of the data samples. A
MEMS GPS is unit to determine positioning and to prevent clock-drift. Due to the small dimension of MEMS,
and their low energy consumption, the "infrasound-logger" is a pocket-size measurement platform, powered
by an 1800 mAh lithium battery. The platform does not require any infrastructure (e.g., data connection,
power supply and specific mounting) like commonly used for the deployment of high-fidelity systems, which



---

makes it mobile and allows rapid deployments as well as measurements at remote places.

The "infrasound-logger" is specifically designed to measure infrasound. The platform hosts the KNMI mini-MB, which is a novel design with a pressure dome as inlet, the casing as backing-volume with a PEEKsil capillary, and the DLVR-F50D as sensing element. The low-frequency cut-off of mini-MB depends on the size of the backing volume, and the characteristics of the capillary. The high-frequency cut-off depends on the inlet parameters of the mini-MB, which is partly controlled by a Gore-Tex air-vent (Section 3.1.4). The "infrasound-logger" has a low-frequency cut-off frequency of $0.044 \pm 0.0025$ Hz, while the high-frequency cut-off varies between 15 and 90 Hz.

A comparison between the mini-MB and a Hyperion infrasound sensor [Merchant, 2015] has shown the differences in amplitude and phase (Figure 4. For the passband frequencies band the mini-MB has an amplitude difference of 30 dB compared to the Hyperion sensor. For the lower frequencies the sensors are in good agreement, both sensors resolved the characteristic microbarom peak around 0.2 Hz [Christie and Campus, 2010]. The higher frequencies, however, show small deviations, which is due to the relatively high noise band of the mini-MB. From 8 Hz onward, the PDF of the mini-MB follows the 12-bit dynamic range of the ADC. Nonetheless, the mini-MB is able to resolve infrasonic ambient noise field up to $\pm$ 8 Hz. Only in case of significant events or extremely noisy conditions, the sensor is capable of sensing pressure perturbations in the higher frequency range.

When the wind noise levels are high, infrasound signals can be masked and remain undetected. Therefore, the sensor platform presents a robust passive anemometer to give insights in the wind conditions during infrasonic measurements. The MEMS anemometer is built up as an omnidirectional sensor. Numerical tests indicate that the temperature difference caused by a wind flow around the thermistors should be significant to be sensed. For validation, the anemometer has been calibrated inside a wind tunnel. Figure 6 shows the outcome of the calibration tests. Based on this outcome, one can conclude that the anemometer can determine wind direction and wind speed, given that the sensor is calibrated. The sensor measures a difference in resistance, which is relative compared to wind speed and direction. Although the sensor is resolving wind direction and speed, the resolution is poor compared to the reference sensors. For the estimation of a 2D gradient (assuming the gradient is uniform), in principle only four degrees of freedom are needed: 2 in the x-direction,





---

2 in the y-direction. Therefore, the proposed system should be over determined in this case. Nonetheless,
the resolution outcome of the MEMS anemometer shows opposite. It is likely that the temperature gradient
is not strong enough to provide a wind direction resolution higher as 30°. A slight deviation is z position (a
height difference) between the thermistors can cause such a reduction of temperature gradient.
Besides an anemometer and infrasound sensor, the platform also hosts a barometric pressure sensor, an
accelerometer, and GPS. Each sensor has been calibrated and compared with a reference sensor. It was
shown that the accelerometer has a relatively high self-noise, which restricts the sensors ability to determine
the ambient seismic noise [Peterson, 1993] [McNamara and Buland, 2004]. Nonetheless, the sensor will most
likely resolve local transient events, which influences the sensitivity of the mini-MB and its ability to resolve
infrasonic sources. The barometric sensor shows good agreement with a reference sensor (Figure 5). Absolute
pressure perturbations due to the weather are resolved. After calibration, the sensor has a precision of $\pm 0.1$
hPa for 93% of the time. For the remainder maximum deviation, compared to the reference sensor, was
$\pm 0.15$ hPa.
Calibration tests, performed in this study and previous literature, show that the MEMS sensors perform
less than the commonly used high-fidelity sensors. The self-noise of the sensors is a critical problem.
Furthermore, the manufacturer of the MEMS sensors highlight there is a significant change of measure-
ment drift [DLV, 2019] [TDK, 2018] [STMicroelectronics, 2017] [STMicroelectronics, 2018], regular calibra-
tion is needed. Nonetheless, the MEMS sensor techniques are continuously developing [Jacob et al., 2014]
[Johari, 2003]. The design of the "infrasound-logger" is such that the platform can be adjust and improved
by adding or swapping sensors. Mobile sensor platforms, build up by PCB's and digital MEMS sensors, are
therefor scalable, flexible, and ready for various geophysical measurements.
Nonetheless, a low-cost mobile multidisciplinary sensor platform can complement existing high-fidelity geo-
physical sensor networks. This study showed that, as long as the MEMS are well-calibrated, they perform
in agreement with the reference sensors. Therefore, the 'infrasound-logger' can contribute significantly to
providing observations during rapid deployments, to complement the existing sensor network by increasing
the number of observations. Although the sensor data does not fully satisfy the measurement requirements,
the improve of spatial resolution enables stacking the observations. This can be realized by stacking the



output of various sensor platforms, or by adding more sensors to the same sensor platform and averaging the output [Nishimura et al., 2019]. Stacking improves the signal-to-noise ratio increases by $1/\sqrt{N}$, where $N$ is the number of observations. Furthermore, the platform enables measurement campaigns at remote places (e.g., weather towers, weather balloons).

# Acknowledgements

The authors thank the calibration lab of the KNMI for their collaboration, explaining the different calibration techniques, and allowing experimental tests at their facility. Furthermore, the authors would like to thank Sam Patrick, Mathieu Basille, Susana Clusella-Trullas, Thomas Clay, Rocio Joo, and Jeff Zeyl for their input regarding the selection of the used MEMS sensors. All figures have been created using Generic Mapping Tools [Wessel et al., 2013]. O.d.O and J.A are funded by a Human Frontier Science Program Young Investigator Grant (SeabirdSound - RGY0072/2017). L.E contribution is funded through a VIDI project from the Dutch Research Council (NWO), project number 864.14.005.

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

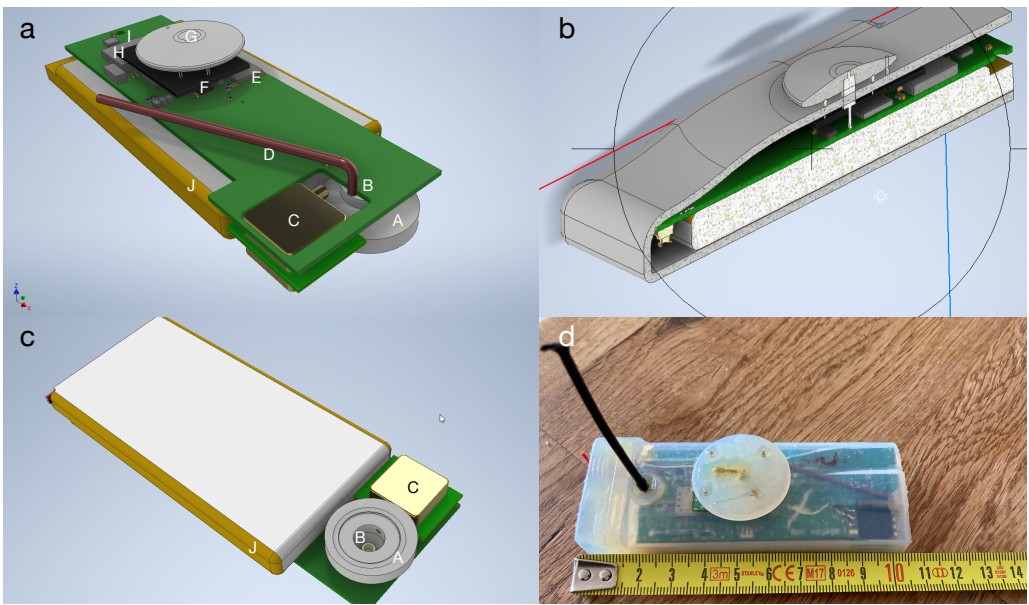

Figure 1: 3D CAD design of (a) the top of the PCB, (b) the casing, (c) the bottom of the PCB with pressure dome, and (d) a picture of the actual platform. The PCB hosts; a pressure dome (a-A/c-A), a barometric pressure sensor (a-B/c-B), a differential pressure sensor (a-C/c-C), a PEEKsil$^{\text{TM}}$ Red series capillary (a-D), an accelerometer (a-F), an anemometer (a-F) with heating element (a-G), a microcontroler (a-H), a GPS (a-I), and a lithium battery (a-J/c-J).



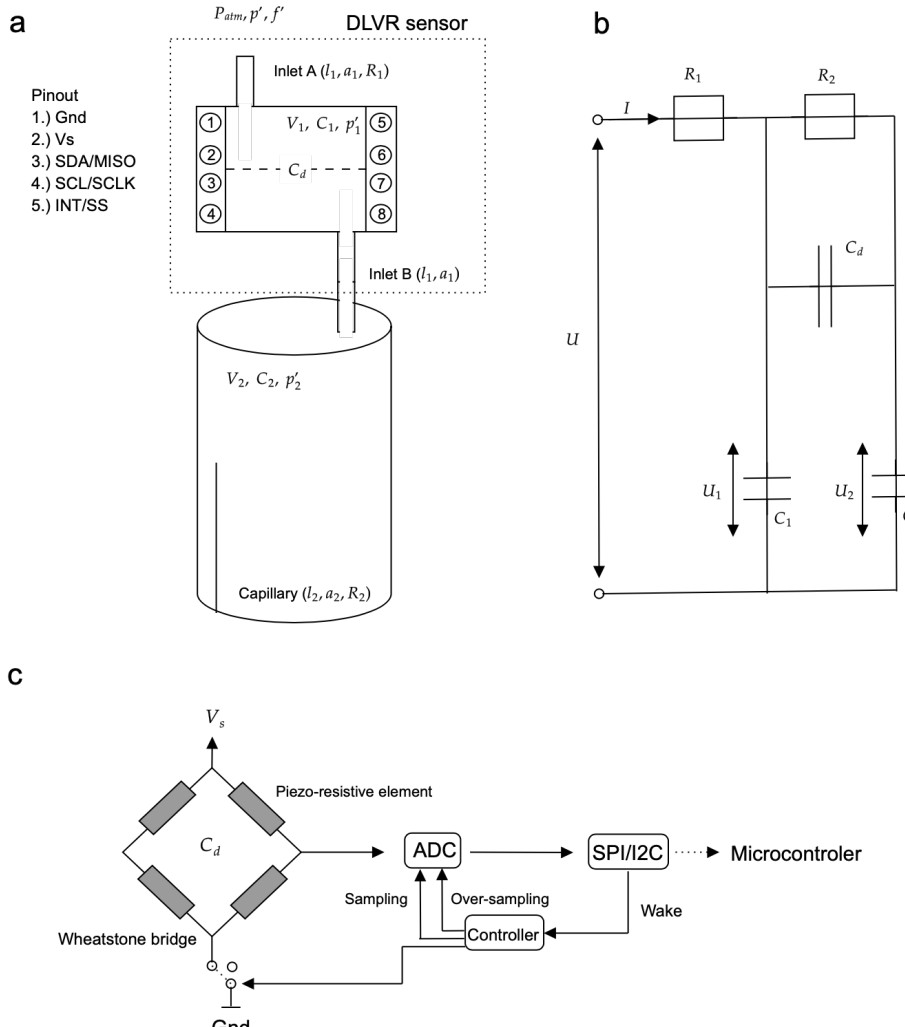

Figure 2: The KNMI mini-MB design with the DLVR sensor and the parameters as listed in Table 1 (a), as well as the electrical circuit of the mini-MB (b). Panel (c) visualises the DLVR sensor.

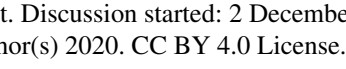

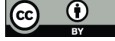

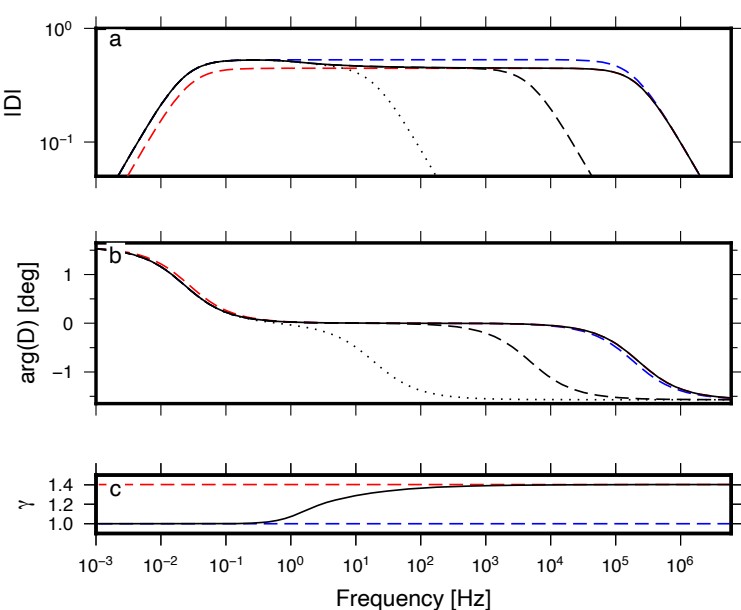

Figure 3: The theoretical sensor frequency response function for (a) amplitude and (b) phase in the case of isothermal and adiabatic gas behaviour in blue and red, respectively. The solid black line indicates the corrected sensor response by $\overline{\gamma}$ (c), as discussed in Section 3.1.3. The dotted and dashed line indicate the shifting high frequency cut-off due to $R_{\mathrm{gore}}$, as discussed in section 3.1.4.



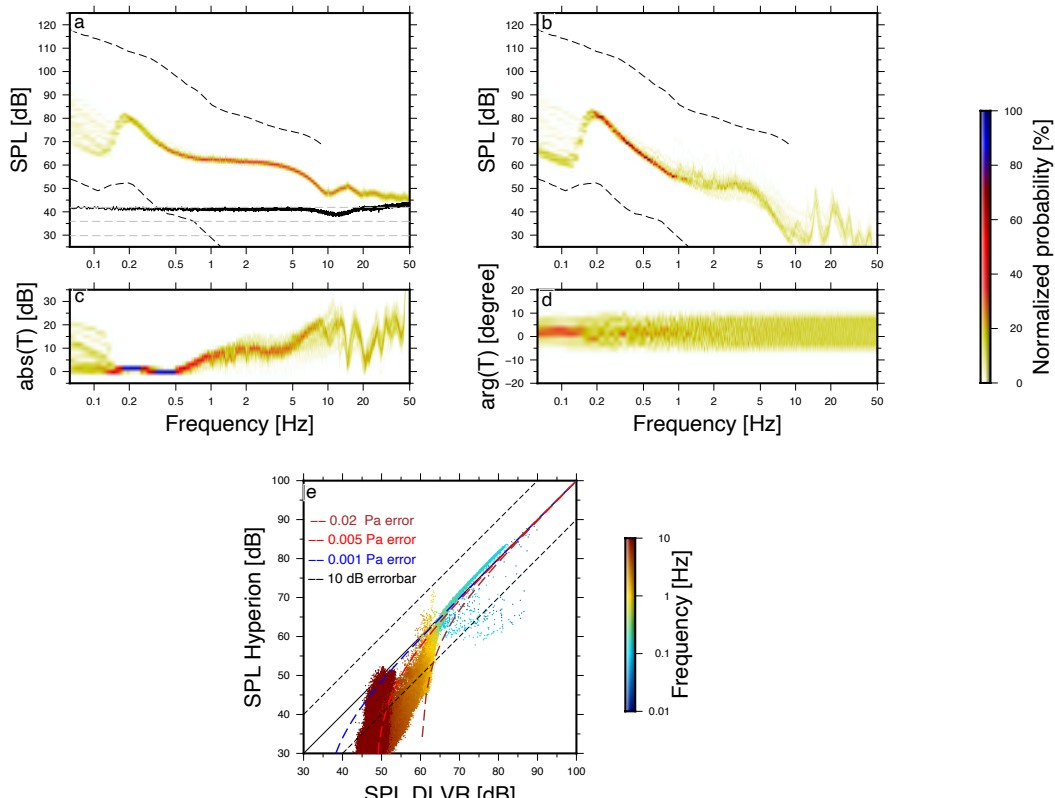

Figure 4: PDF's of pressure spectra recorded with the mini-MB (a) and the Hyperion sensor (b) for a week of continuous recording in dB re. $20^{-6}$ Pa$^2$/Hz. The dotted lines indicate the infrasonic high and low ambient noise levels. Panel (a) shows as well the PSD of the 24hr self-noise recording of the mini-MB in black. Panels (c) and (d) visualise the absolute difference T in amplitude and phase between the mini-MB and the Hyperion as a function of frequency. Panel (e) displays the differences in sound pressure level measured by the mini-MB and the Hyperion sensor for the various frequencies.

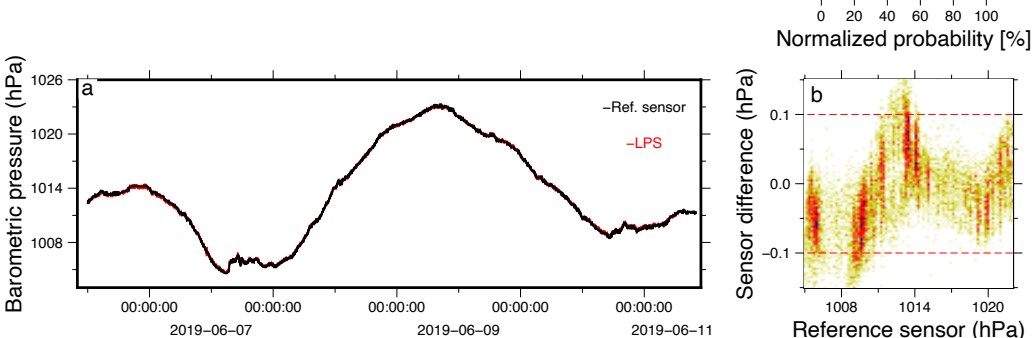

Figure 5: A comparison between the Barometric MEMS sensor (red) and a KNMI reference barometer (black).

Panel (a) shows five days of barometric pressure recordings using both sensor, while panel (b) displays the

difference in measured barometric pressure by the MEMS and the reference sensor.

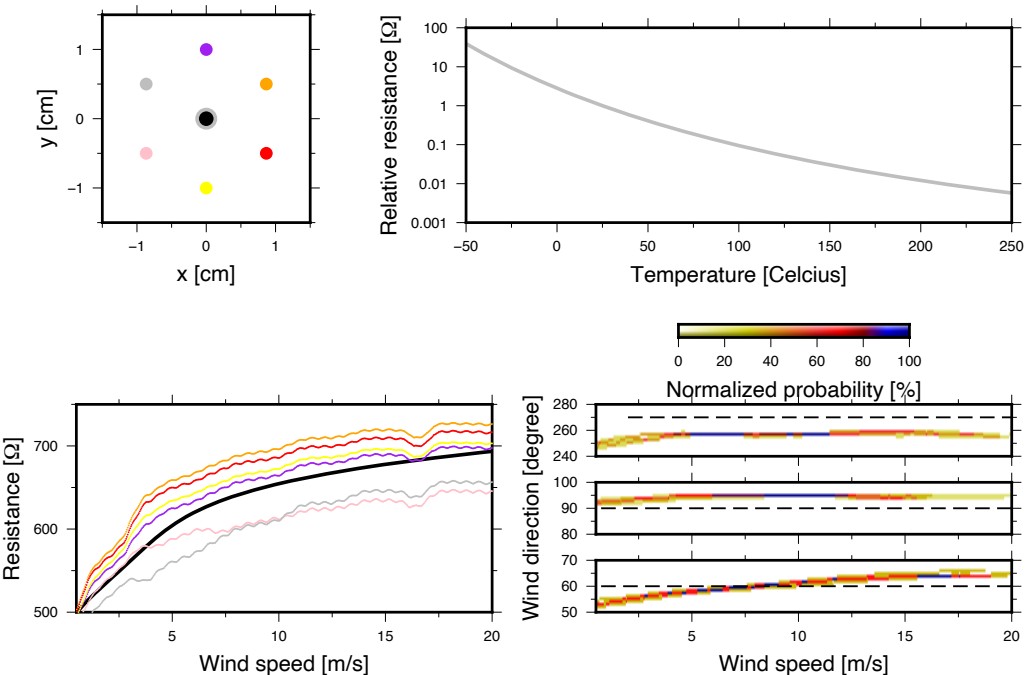

Figure 6: Analyses of the anemometer. Panel a shows the top view of the sensor design, with the central

heating element. Panel b indicates the resistivity of the thermistors over temperature. The measured resis-

tance of the thermistors for calibration set-up a, the colors are in agreement with the sensor design (a), are

shown in panel c. The solid black line is the average 4th order polynomial fit. Panel d indicates the resolved

wind direction (solid lines) compared with the expected direction (dotted lines) of set-ups a, b, and c.



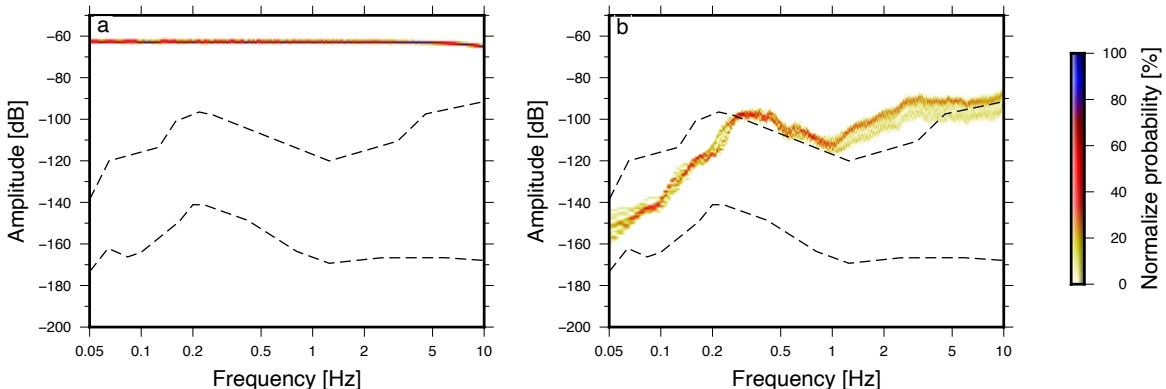

Figure 7: PDF's of the LSM IMU accelerometer (a) and the Streckeisen STS-2 connected to a Quanterra Q330 (b) for 24 hours of continuous recording in dB re. $m^2 s^{-4} Hz^{-1}$. The dotted lines indicate the seismic high and low ambient noise levels.