# Peer review of "The INFRA-EAR: a low-cost mobile multidisciplinary measurement platform for monitoring geophysical parameters"

_Atmospheric Measurement Techniques, 2020_

## Referee Comment (RC1) · Anonymous Referee #2 · 28 Dec 2020

The authors describe a new geophysical sensor package that focuses on infrasound. The unit is remarkably small, lightweight, low power, and ideal for temporary or remote deployments. The authors suggest it could be used in mobile platforms as well - balloons and perhaps oceangoing vessels are implied. The work describes a set of detailed tests on each sensor in the package, as well as theoretical calculations describing the expected response. The authors conclude with a discussion of the strengths and weaknesses of the package compared to other extant solutions.

This is good work and worthy of publication after some further background work and motivation. The technical content appears sound, and the device is well character-

ized. It joins a bevy of low cost infrasound sensor/logger combinations, such as the Gem (cited, but not specifically mentioned), the Raspberry Boom (not mentioned), and the one discussed in Grangeon & Lesage (DOI: 10.1016/j.jvolgeores.2019.106668, not mentioned). Some discussion on how this particular device differs from them is warranted; see comments below.

MAJOR COMMENTS

1. This paper is similar in scope and intent to Anderson et al, 2018: "The Gem infrasound logger and custom-built instrumentation" and Grangeon and Lesage, 2019: "A robust, low-cost and well-calibrated infrasound sensor for volcano monitoring". The present work includes several other sensors, including accelerometers and anemometers, that the above units lack. This should be highlighted. The authors should also read both of the above papers carefully and specifically address how their unit is different. The Raspberry Boom (Raspberry Pi based infrasound monitor) should also be mentioned.

2. The use cases of the device are not well defined. The Gem unit and the one Grangeon and Lesage developed were originally meant for volcanoes. Is that (one of) the use case(s) envisioned here? The connection between ground motion and infrasound sensor interference is important, but will only be a problem when ground shaking is especially strong. That is, for infrasound studies involving local earthquakes or other strong motion sources (see Johnson et al (2020) "Mapping the sources of proximal infrasound" or Bowman, 2019 "Yield and emplacement depth effects on acoustic signals from buried explosions in hard rock".). Maritime environments are mentioned and might make a very good fit, I suggest the authors look up the chapter by Grimmett et al. in the second volume of Infrasound Monitoring for Atmospheric Studies. Balloons are also mentioned – the recent article by Poler and others is cited. The sensor noise level of 0.05 Pa is generally too high for ambient infrasound studies on stratospheric balloons, although focused efforts against loud targets (ground explosions, the microbarom) might be possible. The inclusion of the accelerometer reminds me of the

recent paper "An active source seismo-acoustic experiment using tethered balloons to validate instrument concepts and modelling tools for atmospheric seismology", which might suggest a better use case.

3. I am very skeptical about the utility of the anemometer. The tests were performed under constant temperature and humidity conditions, but it seems to me that different ambient temperatures would really affect its performance. While knowing the wind speed is indeed useful for assessing the source of infrasound background noise, it is generally very clear when interference is due to wind or other sources. Finally, I am not clear why the wind direction is relevant.

4. My general complaint with this type of paper is that the sensor availability is not described. How can the scientific audience get their hands on one of these devices? Will they be ever available for sale, or perhaps part of an equipment pool? This is very important information for groups that may be weighing the option of developing their own units vs. purchasing those already made by others.

MINOR COMMENTS:

Line 13: I suggest a less generic name. Infrasound loggers already exist. Something clever and memorable would be nice here. Lines 51-58: I don't think the discussion of citizen weather stations is particularly relevant. Lines 59-60: Be specific here – e. g. on buoys in the open ocean (cite Grimmett) and on stratospheric balloons (cite Poler). 75: its, not it's 76-77: Here is where a paragraph comparing the unit with others such as the Gem, Raspberry Boom, etc., would be very useful 91: Integrating with existing sensor infrastructures is repeated throughout the paper but no examples of how this would be done are given 94: It is not "novel", there are similar sensor packages already available (e. g. the Gem). 105: How many days can it run on one battery charge? 115: mb or gb? 133: List horizontal and vertical accuracy, and whether it can function above 60,000 ft. This is important if the unit is deployed on a balloon. 147: Sensor self noise is seldom a problem on surface instruments, but it is a major problem on balloon-borne

microbarometers. 149 A comma missing here? 161-166 Particle velocity sensors are pretty rare and probably not worth mentioning, especially since the present unit doesn't use them. 167-169: But this is not true on balloons, see spectra in Bowman and Albert (2018) "Acoustic Event Location and Background Noise Characterization on a Free Flying Infrasound Sensor Network in the Stratosphere" 188: Isn't this the same sensor, or at least very similar, to the one used by Gems, InfraBSUs, and the Raspberry Boom? 282: How were these resistivity values determined? From the manufacturer? 373-375: In general wind noise is pretty obvious from the infrasound time series itself, and the added effort of an anemometer may not be strictly necessary in many cases. Also, how will the anemometer work in extreme environments, such as maritime or high altitude applications? 397: What is ANSYS? 402-406: Can wind speed and direction be accurately determined across the whole range of temperature and humidity conditions the sensor is expected to encounter? This is a very specific and relatively benign set of conditions for the test! 431-434: Has the acceleration response of the MEMS microbarometer been investigated? Some MEMS-based infrasound sensors, like the InfraBSU, are remarkably insensitive to acceleration. 489: I would not characterize the anemometer as "robust" since I am not convinced it has been sufficiently tested under the variety of environments it may encounter in the field. 528: A "weather balloon" is a specific term for a continuously ascending latex balloon carrying a radiosonde. If a long duration drifting balloon like the one described by Poler is intended, please use the term "scientific balloon". Figure 4: If the IMS curves are being used for reference, please make that clear and cite Brown et. al (2014) Figure 7: Please also cite the source of these noise models.

---

## Referee Comment (RC2) · Anonymous Referee #1 · 30 Dec 2020

**Summary:**

This paper describes the design, calibration results, and initial testing of a small instrument that has the capability to measure multiple variables, with an emphasis on infrasound. Overall, the manuscript was well-written and easy to read, though there are a few typos throughout (some are pointed out below). The subject matter is generally appropriate for the AMT journal and the topic is practical and interesting. Since this work is being presented to an atmospheric-leaning audience, I have a few suggestions in the "General Comments" below which I think should be addressed prior to publication of this work. It seems like some of the measurements are not quite as high-

quality as one would like (especially the wind sensor, see comments below); however, I appreciate the honest assessment of the measurements by the authors.

**General Comments:**

1. In the title, the words "geophysical parameters" are vague. Since the centerpiece of the instrument package is the infrasound portion, it seems like it would be appropriate to have "infrasound" included in the title. Something like:

"A low-cost mobile multidisciplinary measurement platform for monitoring infrasound"

2. Since you have submitted this to a journal which emphasizes atmospheric measurements, it seems appropriate to have some discussion about the inlet port used to obtain the (static) pressure. Though the wind/turbulence in this study is considered "wind-noise" (i.e., p.3, l.80; p.15, l.345-346; p.16, l.363-365), there has been quite a bit of work on static-pressure inlet ports which are not mentioned or considered. Perhaps this is a case where "one's persons noise, is another persons signal"; however, I think that the so-called noise is primarily due to dynamic-pressure effects on the port where the pressure is sensed. In the atmospheric community this has typically been dealt with by using a port which reduces the effect of dynamic pressure on the sensed static pressure. such as the Nishiyama-Bedard quad-disk. For examples, see work by Nishayama, 1991; Wilczak, 2004; and Zhang 2011. There is also a paper in review by Burns 2021 (which may not yet be available), but has related information. For example, the inlet port would be an important consideration, when the sensor is deployed on a tower. I would appreciate some comment and/or insight into whether the inlet port is considered important (or not) for the infrasound-logger.

3. In the atmospheric flux community, 3D wind is usually measured with sonic anemometers. Some of these have become quite small, e.g., the TriSonica:

https://www.apptech.com/products/ultrasonic-anemometers/trisonica-mini/

Was this type of technology ever considered for measuring wind with the infrasound logger? This could eliminate the need to generate heat to measure the wind. Also, to deploy the wind sensor on the infrasound-logger means the entire instrument/enclosure needs to be mounted outside at the location where the wind is measured—is that correct? If so, does the box itself present an issue due to distortion of the wind? The ability to displace the wind sensing element away from the infrasound logger box has some practical advantages (and it's unclear if this is possible with the current setup). To convince me that the wind sensor is actually useful, I think a data comparison between the infrasound logger wind speed and direction with a standard wind sensor (in the real atmosphere, outside of a wind tunnel) should be included in the manuscript.

4. The infrasound logger has 64 mb flash memory for data storage (p.5, l.115). What is the typical sample rate used to collect data (based on Fig. 4, looks to be around 100 samples/sec)?....how long can it run unattended without filling up the 64 mb flash memory? Are there communication capabilities (e.g., WIFI, network port, etc)? How do you get data off of it? Is there any custom software (which language) used to make everything work? Can some of these details be described?

**Specific Comments:**

* was the EGU journal, "Geoscientific Instrumentation, Methods and Data Systems" considered as a publication option?

* p.2, l.54, "...short-term and now-casting weather forecasts." include a reference?

* p.5, l.108, "...either be done..." fix typo.

* p.5, l.113, "build-in" should be built-in. "bite" should be either "byte" or "bit"?

* p.15, l.329, what is a "high-frequency shroud"? Is there a reason you need an acronym (HF?) for it? Is it only on the Hyperion sensor inlet and not the mini-MB inlet?

* p.15, l.337-338, why does a bias +/- deviations in dB convert to something that has

only +/- deviations in Pa?

* p.16, l.366, the -5/3 slope is not really "noise", it is related the cascade of turbulent energy (see George, 1984; Zhang, 2011 for details).

* p.14, l.321, p.15, l.341; I don't quite follow what the 12-bit dynamic range effects on the high-freq spectra are....comparing Fig 4a and 4b, the peaks in the Hyperion spectra for f >10 Hz) are due to the limits of the 14-bit ADC on the mini-MB? If a 24-bit ADC was used with the mini-MB would it fix this issue? What is the cause of the high-freq peaks in the Hyperion spectra? Are these real infrasound phenomena that the mini-MB is missing?

* p.16, l.348, it was mentioned a few times that (air) temperature is important, but the sensor does not measure this (or humidity). These seem like important atmospheric variables that are missing from the sensor package...

* p.18, l.297, define ANSYS?

* p.18, l.400, The atmosphere is turbulent. It sounds like this is an issue.

* p.18, l.417, "..different angles with respect to the air flow." Does this mean the yaw angle was varied? What about the pitch angle?

* p.21, l.480, "..phase (Figure 4." missing closing parenthesis.

* p.22, l.516, "adjust" should be "adjusted".

* Fig. 4, the caption states, "dotted lines", but do you mean dashed lines? Also, in panel (a), the horizontal gray dashed lines should be explained.

* Many words in the references need capital letters (needs to be fixed)

**Related References:**

Bedard, A. J., Georges, T. M., 2000. Atmospheric infrasound. Physics Today 53, 32-37. doi:10.1063/1.883019.

Burns, S. P., Frank, J. M., Massman, W. J., Patton, E. G., and Blanken, P. D., 2021. The effect of static pressure-wind covariance on vertical carbon dioxide exchange at a windy subalpine forest site. Agricultural and Forest Meteorology, in-review.

George, W.K., Beuther, P.D., Arndt, R.E.A., 1984. Pressure spectra in turbulent free shear flows. J. Fluid Mech. 148, 155-191. doi:10.1017/S0022112084002299

Nishiyama, R. T., Bedard, A. J., 1991. A "Quad-Disk" static pressure probe for measurement in adverse atmospheres: With a comparative review of static pressure probe designs. Rev. Sci. Instrum. 62, 2193-2204. doi:10.1063/1.1142337.

Wilczak, J. M., Bedard, A. J., 2004. A new turbulence micro-barometer and its evaluation using the budget of horizontal heat flux. J. Atmos. Oceanic Technol. 21, 1170-1181.

Zhang, J., Lee, X., Song, G., Han, S., 2011. Pressure correction to the long-term measurement of carbon dioxide flux. Agric. For. Meteor. 151, 70-77. doi:10.1016/j.agrformet.2010.09.004.
* * *

---

## Author Comment (AC1) · 16 Feb 2021

This paper describes the design, calibration results, and initial testing of a small instrument that has the capability to measure multiple variables, with an emphasis on infrasound. Overall, the manuscript was well-written and easy to read, though there are a few typos throughout (some are pointed out below). The subject matter is generally appropriate for the AMT journal and the topic is practical and interesting. Since this work is being presented to an atmospheric-leaning audience, I have a few suggestions in the "General Comments" below which I think should be addressed prior to publication of this work. It seems like some of the measurements are not quite

as high quality as one would like (especially the wind sensor, see comments below); however, I appreciate the honest assessment of the measurements by the authors.

We would like to thank the reviewer for its careful, positive, and constructive review of our paper, and have included our responses towards your comments in this response letter. Changes to the manuscript based on the comments have been made and highlighted in a marked-up version of the manuscript. The comments have really helped us to produce a much improved manuscript and we thank you for your diligence and attention to detail.

General Comments:
1. In the title, the words "geophysical parameters" are vague. Since the centerpiece of the instrument package is the infrasound portion, it seems like it would be appropriate to have "infrasound" included in the title. Something like: "A low-cost mobile multidisciplinary measurement platform for monitoring infrasound"

The logger has been designed as a multidisciplinary sensor platform. However, the KNMI mini-MB is not an 'off-the-shelf' MEMS sensor and can not be bought online as described within the paper. Therefore, an extra introduction and explanation have been added to the mini-MB. Reviewer 2 asked for a different name for the device, which has now been changed from 'infrasound-logger' to 'INFRA-EAR' (Infrasound and Environmental Atmospheric data Recorder). The title, therefore, has been modified to: 'The INFRA-EAR: a low-cost mobile multidisciplinary measurement platform for monitoring geophysical parameters.'

2. Since you have submitted this to a journal which emphasizes atmospheric measurements, it seems appropriate to have some discussion about the inlet port used to

obtain the (static) pressure. Though the wind/turbulence in this study is considered "wind-noise" (i.e., p.3, l.80; p.15, l.345-346; p.16, l.363-365), there has been quite a bit of work on static-pressure inlet ports which are not mentioned or considered. Perhaps this is a case where "one's persons noise, is another persons signal"; however, I think that the so-called noise is primarily due to dynamic-pressure effects on the port where the pressure is sensed. In the atmospheric community this has typically been dealt with by using a port which reduces the effect of dynamic pressure on the sensed static pressure. such as the Nishiyama-Bedard quad-disk. For examples, see work by Nishayama, 1991; Wilczak, 2004; and Zhang 2011. There is also a paper in review by Burns 2021 (which may not yet be available), but has related information. For example, the inlet port would be an important consideration, when the sensor is deployed on a tower. I would appreciate some comment and/or insight into whether the inlet port is considered important (or not) for the infrasound-logger.

Air turbulence can generate dynamic pressure effects or stagnation pressure at the pressure dome [Raspet et al.,2019]. The stagnation pressure increases with altitude, which results in higher wind speeds. Atmospheric measurements at altitude might therefore be influenced by stagnation pressure [e.g., Bowman et al., 2015, Smink et al., 2020, Krishnamoorthy et al., 2020]. The influence of stagnation pressure on pressure measurements is theoretically elucidated by [Raspet and Webster 2008].
The application of a quad-disk might remove the stagnation pressure. Quad-disks are developed to cancel dynamic pressure effects, and helps detect slower static pressure changes or acoustic perturbations. Theoretical analysis of the quad-disk indicates that it should remove sufficient dynamic pressure to be useful for turbulence studies [Wyngaard et al., 1994]. However, recent studies have shown a minimum effect of quad-disks on infrasound recordings [Krishnamoorthy et al., 2020]. The casing of the INFRA-EAR is designed and developed for mobile and rapid deployments at remote places, adding a quad-disk to the design will expand the dimensions of the casing. Moreover, the pressure dome is positioned at the bottom of the casing, not orientated

towards the dominant wind direction, in order to minimise the stagnation pressure on the pressure sensors.

The comparison between the barometric pressure sensor and reference sensor (Figure 5) does not show a lack of resolution within the barometric pressure recording. The infrasound calibration has been done within a shelter, which lowers the possibility of stagnation pressure on the sensor. Atmospheric tower measurements are one of the use cases for the INFRA-EAR. The application of quad-disks as an inlet port for the INFRA-EAR pressure sensors is of interest for measurements around the atmospheric boundary layer and needs further review. This explanation has been added to the manuscript in section 2.2.

3. In the atmospheric flux community, 3D wind is usually measured with sonic anemometers. Some of these have become quite small, e.g., the TriSonica: https://www.apptech.com/products/ultrasonic-anemometers/trisonica-mini/ Was this type of technology ever considered for measuring wind with the infrasound logger? This could eliminate the need to generate heat to measure the wind. Also, to deploy the wind sensor on the infrasound-logger means the entire instrument/enclosure needs to be mounted outside at the location where the wind is measured—is that correct? If so, does the box itself present an issue due to distortion of the wind? The ability to displace the wind sensing element away from the infrasound logger box has some practical advantages (and it's unclear if this is possible with the current setup). To convince me that the wind sensor is actually useful, I think a data comparison between the infrasound logger wind speed and direction with a standard wind sensor (in the real atmosphere, outside of a wind tunnel) should be included in the manuscript.

The sensor-platform as described in the paper is one unit, that is to say all the sensors are physically connected to the PCB. The anemometer, therefore, can not be unmounted from the PCB and be used separately. This indeed means that the entire

enclosure needs to be mounted outside. The casing has been designed to provide laminar wind flow around the anemometer, avoiding turbulence around the sensing elements.

Furthermore, the casing and sensor-platform has been designed to be firm and robust, so the sensor can function under harsh environmental conditions.

In total 25 INFRA-EAR loggers have been produced and deployed during a 2020 field campaign at Crozet Island in the Southern Ocean. These loggers have been fitted to Wandering Albatrosses as bio-loggers. As this was foreseen, the loggers had been designed to be able to withstand extreme conditions (e.g., extreme weather, damage by hits/beaks, and diving). A mechanical anemometer, therefore, was discarded as such devices would easily damage underwater. A sonic anemometer requires a relatively high power supply (see remark 4). Instead, it was opted to design an anemometer that is inspired by a 2D hot-wire anemometer, i.e. a passive anemometer. Within the paper we show how the anemometer functions within a controlled calibration area (i.e., a wind tunnel). In the revised manuscript, we have added and modified the analyses of the calibration and have expressed the shortcomings in more detail. The analysis of the anemometer has been expressed at line 416, which shows how to convert thermistor measurements into a numerical temperature gradient. The gradient is used to determine the wind speed and direction. This approach improves the analyses. Furthermore, it enables to add a statistical error analysis to the measurements. Future studies will focus on the analysis of the 2020 field season data and will discuss how the anemometer compares to model data.

The remarks about the anemometer have been considered within the paper at line 416. Future 2D hot-wire anemometers should be considered with a minimum of 8 thermistors, in order to exclude geometric uncertainties (Line 442).

4. The infrasound logger has 64 mb flash memory for data storage (p.5, l.115). What is the typical sample rate used to collect data (based on Fig. 4, looks to be around 100 samples/sec)?....how long can it run unattended without filling up the 64 mb flash

memory? Are there communication capabilities (e.g., WIFI, network port, etc)? How do you get data off of it? Is there any custom software (which language) used to make everything work? Can some of these details be described?

The PCB sensors are controlled by an MSP430 microcontroller, which is integrated on the PCB and is powered by an 1800 mAh lithium battery. The microcontroller runs on self-made software. The microcontroller defines the sample time, sample frequency, and data storage. For Fig.4, a sampling frequency of 100 Hz has been used. The limitation of the platform is not the flash memory but its battery life. The platform, concurrently measuring various geophysical parameters, requires a specific power supply, emptying the battery before running out of memory.

Moreover, no data processing is performed by the microcontroller. The data is stored digitally. The casing functions as protection and as a backing volume for the differential pressure sensor. The battery has therefore been included inside the casing. To extract data from the platform, the platform needs to be connected to a computer. There are (yet) no wireless communication possibilities.

The remarks about the memory and data storage has been taken into account at line 123

Specific Comments:
Was the EGU journal, "Geoscientific Instrumentation, Methods and Data Systems" considered as a publication option?
No, only Atmospheric Measurement Technique has been considered as a publication option.
p.2, l.54, "...short-term and now-casting weather forecasts." include a reference?
Added
p.5, l.108, "...either be done..." fix typo.
Corrected

p.5, l.113, "build-in" should be built-in. "bite" should be either "byte" or "bit"?
Corrected

p.15, l.329, what is a "high-frequency shroud"? Is there a reason you need an acronym (HF?) for it? Is it only on the Hyperion sensor inlet and not the mini-MB inlet?
No need for acronyms, has been removed. Yes the shroud only applies the Hyperion sensor, since this is part of the sensor design.

p.15, l.337-338, why does a bias +/- deviations in dB convert to something that has only +/- deviations in Pa?
Correct. The conversion of the recordings in Pascal to decibels is a logarithmic conversion (i.e., $SPL = 10 * log10(P/(P_{ref}^2))$). The confidence interval in dB indicates the spectra's error, whereas the confidence interval in Pa shows the measurement error of the recordings. It is correct that an error in dB can not be transformed into a negative error in Pa. The sentences are not informative and have been changed.

p.16, l.366, the -5/3 slope is not really "noise", it is related the cascade of turbulent energy (see George, 1984; Zhang, 2011 for details).
Agree, correct.

p.14, l.321, p.15, l.341; I don't quite follow what the 12-bit dynamic range effects on the high-freq spectra are....comparing Fig 4a and 4b, the peaks in the Hyperion spectra for f >10 Hz) are due to the limits of the 14-bit ADC on the mini-MB? If a 24-bit ADC was used with the mini-MB would it fix this issue? What is the cause of the high-freq peaks in the Hyperion spectra? Are these real infrasound phenomena that the mini-MB is missing?
Yes, when a 24bit ADC has been used, the KNMI mini-MB's self-noise level would be lower. Sleeman, et al. 2007 show how the self-noise of an ADC depends on sample frequency and the number of ADC bits. Following this method, the self-noise of the 14-bit ADC of the KNMI mini-MB has been determined. Fig 4a shows the theoretical self-noise level of the ADC (dotted lines indicate 12, 13, and 14 bit ADCs theoretical self-noise levels), the recorded self-noise (solid black lines), and shows that the PDF follows from f>1Hz the theoretical self-noise levels.

We expect that the spectral peaks above 10 Hz correspond to resonances that exist inside the measurement shelter. It is expected that the Hyperion sensor records these pressure fluctuations correctly as the sensor has been calibrated by the vendor. The high-frequency peaks are below the KNMI mini-MB ADC's self-noise levels, and therefore not resolved by the KNMI mini-MB.

p.16, l.348, it was mentioned a few times that (air) temperature is important, but the sensor does not measure this (or humidity). These seem like important atmospheric variables that are missing from the sensor package...

Correct, both temperature and humidity are interesting parameters to add to the sensor platform. The barometers do measure temperature. However, we doubt the accuracy since those sensors are primarily built to measure the barometric pressure. Future platforms should include such sensors.

p.18, l.297, define ANSYS? *

Corrected. ANSYS is a numerical modelling software.

p.18, l.400, The atmosphere is turbulent. It sounds like this is an issue.

The anemometer elements are placed within a couple of millimetres from each other. To make sure the flow passing the anemometer is the atmospheric wind flow, and not changed by the anemometer design or the casing, the casing has been designed to not change the 'initial' wind flow.

p.18, l.417, "..different angles with respect to the air flow." Does this mean the yaw angle was varied? What about the pitch angle?

The calibration measurements have been performed within the horizontal plane. Thus the pitch angle has been zero. This has been added to the text.

p.21, l.480, "..phase (Figure 4." missing closing parenthesis.

Corrected

p.22, l.516, "adjust" should be "adjusted".

Corrected

Fig. 4, the caption states, "dotted lines", but do you mean dashed lines? Also, in panel (a), the horizontal gray dashed lines should be explained.

Corrected. The horizontal dashed line indicates the theoretical sensor self-noise (section 3.1.6)

* Many words in the references need capital letters (needs to be fixed)

Corrected

---

## Author Comment (AC2) · 16 Feb 2021

The authors describe a new geophysical sensor package that focuses on infrasound. The unit is remarkably small, lightweight, low power, and ideal for temporary or remote deployments. The authors suggest it could be used in mobile platforms as well - balloons and perhaps oceangoing vessels are implied. The work describes a set of detailed tests on each sensor in the package, as well as theoretical calculations describing the expected response. The authors conclude with a discussion of the strengths and weaknesses of the package compared to other extant solutions. This is good work and worthy of publication after some further background work and

motivation. The technical content appears sound, and the device is well characterized. It joins a bevy of low cost infrasound sensor/logger combinations, such as the Gem (cited, but not specifically mentioned), the Raspberry Boom (not mentioned), and the one discussed in Grangeon Lesage (DOI: 10.1016/j.jvolgeores.2019.106668, not mentioned). Some discussion on how this particular device differs from them is warranted; see comments below.

We would like to thank the reviewer for its careful, positive, and constructive review of our paper, and have included our responses towards your comments in this response letter. Changes to the manuscript based on the comments have been made and highlighted in a marked-up version of the manuscript. The comments have really helped us to produce a much improved manuscript and we thank you for your diligence and attention to detail.

MAJOR COMMENTS
1. This paper is similar in scope and intent to Anderson et al, 2018: "The Gem infrasound logger and custom-built instrumentation" and Grangeon and Lesage, 2019: "A robust, low-cost and well-calibrated infrasound sensor for volcano monitoring". The present work includes several other sensors, including accelerometers and anemometers, that the above units lack. This should be highlighted. The authors should also read both of the above papers carefully and specifically address how their unit is different. The Raspberry Boom (Raspberry Pi based infrasound monitor) should also be mentioned.
Agree, an extra paragraph is added to the introduction, which highlights earlier work. This has been added around line 78.

2. The use cases of the device are not well defined. The Gem unit and the one Grangeon and Lesage developed were originally meant for volcanoes. Is that (one

of) the use case(s) envisioned here? The connection between ground motion and infrasound sensor interference is important, but will only be a problem when ground shaking is especially strong. That is, for infrasound studies involving local earthquakes or other strong motion sources (see Johnson et al (2020) "Mapping the sources of proximal infrasound" or Bowman, 2019 "Yield and emplacement depth effects on acoustic signals from buried explosions in hard rock".). Maritime environments are mentioned and might make a very good fit, I suggest the authors look up the chapter by Grimmett et al. in the second volume of Infrasound Monitoring for Atmospheric Studies. Balloons are also mentioned – the recent article by Poler and others is cited. The sensor noise level of 0.05 Pa is generally too high for ambient infrasound studies on stratospheric balloons, although focused efforts against loud targets (ground explosions, the microbarom) might be possible. The inclusion of the accelerometer reminds me of the recent paper "An active source seismo-acoustic experiment using tethered balloons to validate instrument concepts and modelling tools for atmospheric seismology", which might suggest a better use case.

The manuscript explains the design, development, and calibration of the INFRA-EAR. Initially, the INFRA-EAR has been designed as a biologger for the monitoring of atmospheric parameters. In total 25 INFRA-EAR loggers are produced and used during the 2020 field campaign at Crozet Island in the Southern Ocean. The loggers are fitted to Wandering Albatrosses as biologgers. The Southern Hemisphere has very little in situ measurements, due to limited shore areas. The use of INFRA-EAR in such areas is ideal. ÂăThe first study of this dataset has been completed and will be submitted soon.
However, thanks to multiple geophysical sensors on the platform, various more applications are possible. The KNMI mini-MB has been designed for infrasonic measurements. Volcanic, earthquake and nuclear monitoring are possible thanks to the mini-MB. Moreover, figure 4 has shown that the mini-MB resolves the microbaroms source peak, which hints that the platform can monitor the infrasonic ambient noise field.
The accelerometer on the platform allows us to monitor strong distant motions, or nearby motions (e.g., earthquake and volcanic monitoring). Furthermore, monitoring of seismo-acoustic events becomes possible thanks to the combination of both the accelerometer and mini-MB.
This extra information about the use cases has been added around line 575.

3. I am very skeptical about the utility of the anemometer. The tests were performed under constant temperature and humidity conditions, but it seems to me that different ambient temperatures would really affect its performance. While knowing the wind speed is indeed useful for assessing the source of infrasound background noise, it is generally very clear when interference is due to wind or other sources. Finally, I am not clear why the wind direction is relevant.

Infrasound array recordings often show the interference of wind, as evidenced by shape of the pressure spectra, revealing the characteristics of turbulence spectra. Therefore, an anemometer is useful to obtain the wind direction and speed, which relate to the local atmospheric infrasound array/station conditions.
In line with Reviewer 1, future platforms should include as well a temperature/humidity sensor. The anemometer on the platform is inspired by a 2D hot-wire anemometer, a passive anemometer. The anemometer analysis has been improved and explained in line 416, which shows how to convert thermistor measurements into a numerical temperature gradient. The gradient is used to determine wind speed and direction. This approach improves the analyses. Furthermore, it enables us to add a statistical error analysis to the measurements. Future 2D hot-wire anemometers should be considered with a minimum of 8 thermistors to exclude geometric uncertainties (Line 442).

[Figure]

4. My general complaint with this type of paper is that the sensor availability is not described. How can the scientific audience get their hands on one of these devices? Will they be ever available for sale, or perhaps part of an equipment pool? This is very important information for groups that may be weighing the option of developing their own units vs. purchasing those already made by others.Âă This paper describes the design, development, and calibration processes of our multidisciplinary sensor platform. Those aspects are of relevant use for various scientific communities. The paper can either function as a guide for calibration of MEMS sensors, design guideline for future sensor-platforms, or as reference for specific MEMS use.

The INFRA-EAR has been used in a scientific field-campaign as biologger. The KNMI mini-MB has been used to resolve infrasonic sources in the Southern Ocean and will be integrated into the sensor network of the KNMI. Furthermore, whoever is interested in this sensor platform can either produce it themself, follow the paper, or contact Dominique Filippi (co-author) of Sextant Technology Inc. Some of the used components can be bought 'off the shelf', and thus the platform could be easily reproduced. In that sense, this paper is also guideline. This has been emphasised in the acknowledgement.

MINOR COMMENTS:
Âă Line 13: I suggest a less generic name. Infrasound loggers already exist. Something clever and memorable would be nice here.
The sensor-platform's name has been changed from 'infrasound-logger' to 'INFRA-EAR', which is an acronym for Infrasound and Environmental Atmospheric data Recorder.
Lines 59-60: Be specific here – e. g. on buoys in the open ocean (cite Grimmett) and on stratospheric balloons (cite Poler).
Corrected

75: its, not it's
Corrected
76-77: Here is where a paragraph comparing the unit with others such as the Gem, Raspberry Boom, etc., would be very useful
Added
91: Integrating with existing sensor infrastructures is repeated throughout the paper but no examples of how this would be done are given
The sensor platform is embedded with digital MEMS, which generate digital outputs. Therefore the outputs can easily be integrated within existing monitoring software. This has been extra emphasis within the introduction.
94: It is not "novel", there are similar sensor packages already available (e. g. the Gem).
It is correct that similar infrasound packages, barometric pressure sensors, anemometers are available. However, the combination of all on one PCB is novel.
105: How many days can it run on one battery charge?
This depends on how many sensors concurrently switch on, how long each sensor measures, and their sample frequency. For example; every hour 5min recording of all sensors, with each measurement a GPS timestamp and a GPS position every 15min, will last approximately 20 days.
115: mb or gb?
mb. The data is stored in bits.
133: List horizontal and vertical accuracy, and whether it can function above 60,000 ft. This is important if the unit is deployed on a balloon.
The GPS's horizontal accuracy is +- 2.5m; for an altitude varying between 0 and 60000ft, this has been added to the text (Line 150). Above 60,000ft (20km) no information about the sensor has been presented by the datasheet/manufacturer, mainly because of the ITAR limits. If the sensor platform can be used on 60,000 ft we have listed the atmosphere's parameters at surface level and 60,000 ft. and compared these values with the operation ranges of the MEMS sensor on the PCB.

The mean surface pressure for 2020 at the KNMI test field (5 E, 52 N) according to ECMWF ERA5 is 1000 hPa, whereas the absolute pressure at 60,000 ft has been specified as 70 hPa. Corresponding to these altitudes, the mean surface temperature and 60,000 ft temperature are 12 and -50, respectively. The relative humidity depends on temperature and pressure, which is $75\%$ and $5\%$ at the surface and 60,000 ft.

The MEMS on the PCB operate according to the specifications listed in the paper, and datasheet. For altitude measurements at 60,000 ft, all MEMS will act/record at the limits according to the datasheets' behaviour. Before a measurement campaign can be executed at 60,000ft, we recommend obtaining knowledge of the sensor behaviour in these circumstances, done by calibration within climate chambers.

A comma missing here?

Corrected

161-166 Particle velocity sensors are pretty rare and probably not worth mentioning, especially since the present unit doesn't use them.

They are indeed very rare. However, differential pressure sensors only provide scalar information about the dynamic pressure field, whereas a particle velocity sensor gives insight in the amplitude and directionality. Such a device could be of interest whenever only one sensor is available (e.g., on a scientific balloon, a biologger). Bringing it up will place it in perspective, therefore we like to keep it within the text.

167-169: But this is not true on balloons, see spectra in Bowman and Albert (2018) "Acoustic Event Location and Background Noise Characterization on a Free Flying Infrasound Sensor Network in the Stratosphere"

The reference of the global noise curves within the text is towards the sensor responses of IMS certificated infrasound sensors, not actual ambient noise recordings. The IMS specifications state that the sensor self-noise should be at least 18 dB below the low noise curves at 1Hz [Brown et al., 2014; Marty, 2019]. This has been added to the text.

188: Isn't this the same sensor, or at least very similar, to the one used by Gems, InfraBSUs, and the Raspberry Boom?

The KNMI mini-MB does not vary extensively from earlier developed differential pressure sensors for infrasound monitoring. However, this is the first miniature differential pressure sensor with an integrated ADC. All the devices listed in the question are analogue MEMS. Moreover, the ability to measure various geophysical parameters concurrently improves the fundamental analyses of infrasound.

This paper has shown how to design and use new sensor techniques to develop miniature sensor-platforms. Integrate various geophysical sensors on one PCB, control the power supply, and divide the measurements in various bursts, which allows concurrently measuring various geophysical parameters on one platform, something the devices within the question cannot do. The PCB is created with the latest technology, including a 3D printed casing.

Furthermore, an extensive analysis regarding the sensors' theoretical responses has been presented, as well as calibration protocols.

282: How were these resistivity values determined? From the manufacturer?

Correct. The citation towards the manufacturer has been added.

373-375: In general wind noise is pretty obvious from the infrasound time series itself, and the added effort of an anemometer may not be strictly necessary in many cases. Also, how will the anemometer work in extreme environments, such as maritime or high altitude applications?

Agree. Those things are interesting to discover and will follow soon after. The anemometers' calibration shows how the anemometer functions and how future measurement should be interpreted (by knowing its behaviour in the controlled area). Having independent measurements of wind is useful for the interpretation of turbulence spectra. Within the INFRA-EAR design as biologger, the anemometer was part of the application requirements. For a different set of design requirements, the anemometer could be omitted.

397: What is ANSYS?

Corrected in the text.

402-406: Can wind speed and direction be accurately determined across the whole range of temperature and humidity conditions the sensor is expected to encounter?

This is a very specific and relatively benign set of conditions for the test!

Correct. Ideally, the calibration conditions are valid for the entire range of temperature and humidity. However, the wind tunnel of the KNMI is not placed inside a climate chamber, so such a calibration test is not possible, and this information has to be obtained during field measurements. Future work will have to address this. Furthermore, future sensor platforms should include a temperature/humidity sensor to measure those atmospheric parameters concurrently. This has been added to the discussion.

431-434: Has the acceleration response of the MEMS microbarometer been investigated? Some MEMS-based infrasound sensors, like the InfraBSU, are remarkably insensitive to acceleration.

Âă No acceleration response has been obtained for the mini-MB. However, the mini-MB is the digital version of the InfraBSU; a similar response is expected.

489: I would not characterize the anemometer as "robust" since I am not convinced it has been sufficiently tested under the variety of environments it may encounter in the field.

Agree, correct

528: A "weather balloon" is a specific term for a continuously ascending latex balloon carrying a radiosonde. If a long duration drifting balloon like the one described by Poler is intended, please use the term "scientific balloon".

Corrected

Figure 4: If the IMS curves are being used for reference, please make that clear and cite Brown et. al (2014)

Corrected

Figure 7: Please also cite the source of these noise models.

Corrected

**Supplement:**

[revised manuscript text omitted]
}10^{-4}\text{m}$ | Thermal conductivity | $\kappa = 2.5\text{x}10^{-2}\text{ W m}^{-1}\text{ K}^{-1}$ |
| Diaphragm sensitivity | $C_d = 7.5\text{x}10^{-11}\text{m}^4\text{s}^2\text{kg}^{-1}$ | Heat capacity | $\rho\, c_p = 1.1\text{x}10^3\text{ J m}^{-3}\text{ K}^{-1}$ |
| **Parameters** | | | |
| Inlet resistance | $R_1 = 8.7\text{x}10^3\text{ kg m}^{-4}\text{ s}^{-1}$ | Fore volume | $V_1 = 4.5\text{x}10^{-7}\text{ m}^3$ |
| Capillary resistance | $R_2 = 2.3\text{x}10^{10}\text{ kg m}^{-4}\text{ s}^{-1}$ | Backing volume | $V_2 = 16.5\text{x}10^{-6}\text{ m}^3$ |
| Size fore volume | $L_1 = 2\text{x}10^{-4}\text{m}$ | Size backing volume | $L_2 = 4\text{x}10^{-4}\text{
[revised manuscript text omitted]